# Non-canonical Hedgehog signaling regulates spinal cord and muscle regeneration in *Xenopus laevis* larvae

**Andrew M Hamilton\*, Olga A Balashova, Laura N Borodinsky\***

Department of Physiology & Membrane Biology Shriners Hospitals for Children Northern California, University of California, Sacramento, School of Medicine, Sacramento, United States

**Abstract** Inducing regeneration in injured spinal cord represents one of modern medicine's greatest challenges. Research from a variety of model organisms indicates that Hedgehog (Hh) signaling may be a useful target to drive regeneration. However, the mechanisms of Hh signaling-mediated tissue regeneration remain unclear. Here, we examined Hh signaling during post-amputation tail regeneration in *Xenopus laevis* larvae. We found that while Smoothened (Smo) activity is essential for proper spinal cord and skeletal muscle regeneration, transcriptional activity of the canonical Hh effector Gli is repressed immediately following amputation, and inhibition of Gli1/2 expression or transcriptional activity has minimal effects on regeneration. In contrast, we demonstrate that protein kinase A is necessary for regeneration of both muscle and spinal cord, in concert with and independent of Smo, respectively, and that its downstream effector CREB is activated in spinal cord following amputation in a Smo-dependent manner. Our findings indicate that non-canonical mechanisms of Hh signaling are necessary for spinal cord and muscle regeneration.

**\*For correspondence:**
andrewmichaelhamilton@gmail.com (AMH);
lnborodinsky@ucdavis.edu (LNB)

**Competing interests:** The authors declare that no competing interests exist.

## Introduction

Injury in low regenerative capacity tissues such as spinal cord represents a persistent challenge in modern medicine, with millions of cases of spinal cord-related disability worldwide (*Lee et al., 2014*). There are currently no therapies for repairing severe damage in the mammalian central nervous system, but a variety of paradigms have shown great promise in diverse model systems, including stem cell transplantation (*Dasari et al., 2014*; *Lu et al., 2012*; *Méndez-Olivos et al., 2017*), pharmacological and genetic manipulation of apoptosis, reactive oxygen species, ion channel function, growth factor and morphogenetic protein signaling, and the immune response (*Akazawa et al., 2004*; *Beck et al., 2003*; *Fukazawa et al., 2009*; *Love et al., 2013*; *Tapia et al., 2017*; *Tseng et al., 2007*; *Tseng et al., 2010*), and direct electrical stimulation (*Gomes-Osman et al., 2016*). Although diverse cellular processes are involved in neural regeneration, one pathway that has consistently proven to be pro-regenerative is Hedgehog (Hh) signaling. Treatment with Sonic hedgehog (Shh) or with agonists of its main effector, Smoothened (Smo), enhances neurological recovery to nerve damage in rats (*Bambakidis et al., 2012*) possibly by enhancing neural cell proliferation (*Bambakidis et al., 2009*), while loss of Shh in zebrafish impairs retinal regeneration (*Sherpa et al., 2014*). Hh signaling is also essential for injury response in a wide variety of tissues, including cardiac muscle (*Kawagishi et al., 2018*; *Singh et al., 2018a*; *Sugimoto et al., 2017*), liver (*Grzelak et al., 2014*), limb (*Singh et al., 2012*; *Yakushiji et al., 2009*), and tail (*Romero et al., 2018*; *Singh et al., 2018b*; *Taniguchi et al., 2014*), making this pathway an attractive therapeutic target.

The Hh pathway is classically known as a primary regulator of organogenesis and tissue homeostasis (*Briscoe and Thérond, 2013*). Hh binding represses its receptor Patched (Ptch), thereby de-represses the G-protein coupled receptor Smo. Canonically, Smo activates the transcription factor Gli2, which drives transcription of both Ptch and the positive feedback regulator Gli1, as well as a wide variety of genes controlling cell migration, differentiation, and especially proliferation (*Briscoe and Thérond, 2013*). In addition to the canonical, Gli-dependent cascade, a number of non-canonical, Gli-independent pathways exist, operating via Src family kinase (*Sloan et al., 2015*; *Yam et al., 2009*), the small GTPases Rac1 and RhoA (*Ho Wei et al., 2018*; *Polizio et al., 2011*), NF-KB activation via PKC (*Qu et al., 2013*), and the $Ca^{2+}$-Ampk axis (*Teperino et al., 2012*), as well as other non-canonical signaling cascades (*Belgacem et al., 2016*; *Teperino et al., 2014*). Our own work has shown that Shh drives spontaneous $Ca^{2+}$ spike activity in immature neurons, regulating neuronal differentiation during spinal cord development (*Belgacem and Borodinsky, 2011*). Moreover, neural tube formation in mouse and frog is associated with a deactivation of canonical Shh signaling (*Balaskas et al., 2012*; *Belgacem and Borodinsky, 2015*; *Lee et al., 1997*), and a concomitant switch to a non-canonical, PKA and $Ca^{2+}$-dependent pathway, which itself contributes to the repression of the canonical, Gli-dependent cascade (*Belgacem and Borodinsky, 2015*).

To better understand its role in tissue repair, we examined the Hh signaling pathway during tail regeneration in *Xenopus laevis* larvae. We found that Hh signaling is necessary for the regeneration of the spinal cord and skeletal muscle, primarily through Gli-independent pathways, and our results implicate PKA/CREB as an interacting signaling cascade. These findings offer the possibility of enhancing regeneration by differentially targeting canonical and non-canonical Hh pathways.

## Results

### Hedgehog signaling regulates regeneration of muscle and spinal cord

*Xenopus* larvae exhibit remarkable regenerative capacity to replenish all the tissues of the tail when amputated. This ability to regenerate the tail's spinal cord, muscle, notochord, and skin is present throughout larval stages (stages 39– 46) and before metamorphosis (stages 49–54), with a refractory period during stages 47–48, during which regeneration is blocked (*Aztekin et al., 2019*; *Aztekin et al., 2020*; *Beck, 2012*; *Ferreira et al., 2018*; *Kakebeen et al., 2020*; *Love et al., 2013*). Both regenerative periods are considered as such, rather than merely developmental growth phases, because during both periods tail amputation leaves a stump containing tissues that have already matured in the larva, unlike the growing tip of the tail that is removed by amputation. Hence, rather than a recapitulation of tail growth, this represents a true regenerative process that requires reactivating mature tissues to replace the missing tail structures.

Our previous studies have shown that electrical activity is important for spinal cord and muscle regeneration in tail-amputated stage 39–41 *X. laevis* larvae (*Tu and Borodinsky, 2014*). In addition, $Ca^{2+}$-dependent activity is necessary for neuronal and muscle cell differentiation during embryonic skeletal muscle and spinal cord development (*Borodinsky et al., 2004*; *Ferrari et al., 1996*; *Ferrari and Spitzer, 1999*; *Gu and Spitzer, 1995*). This activity interacts with non-canonical, $Ca^{2+}$-dependent Shh signaling during spinal cord neuron differentiation (*Belgacem and Borodinsky, 2011*), positioning non-canonical Hh signaling as an excellent candidate for regulating regeneration in muscle and neural tissues. Furthermore, Hh isoforms are expressed during these larval stages in several tissues, including somites, floor plate, and notochord (*Ekker et al., 1995*; *Howell et al., 2002*; *Koide et al., 2006*; *Yin et al., 2010*) and transcripts for Hh ligands are detected through single-cell RNA seq in notochord, floor plate, and various subsets of spinal cord cells in stumps and regenerating tails of stage 40 larvae from 0 through 3 days post amputation (*Aztekin et al., 2019*). We also find Hh protein during these stages in the axial musculature and spinal cord, and the protein level increases after 24 h post amputation (hpa) in the regenerating tail (*Figure 1—figure supplement 1*). Therefore, to directly examine the role of Hh-Smo signaling in tissue regeneration, tails of *X. laevis* larvae were amputated and larvae allowed to regenerate in Smo antagonist cyclopamine, Smo agonist SAG, or vehicle control solution, then fixed at 72 hpa and stained for mitotic activity (phospho-histone H3 [P-H3]), spinal cord (Sox2+ neural stem cells [NSCs] lining the spinal cord central canal), and skeletal muscle (12/101+ differentiated skeletal muscle cells) in the regenerate and vicinity (*Figure 1A–C*).

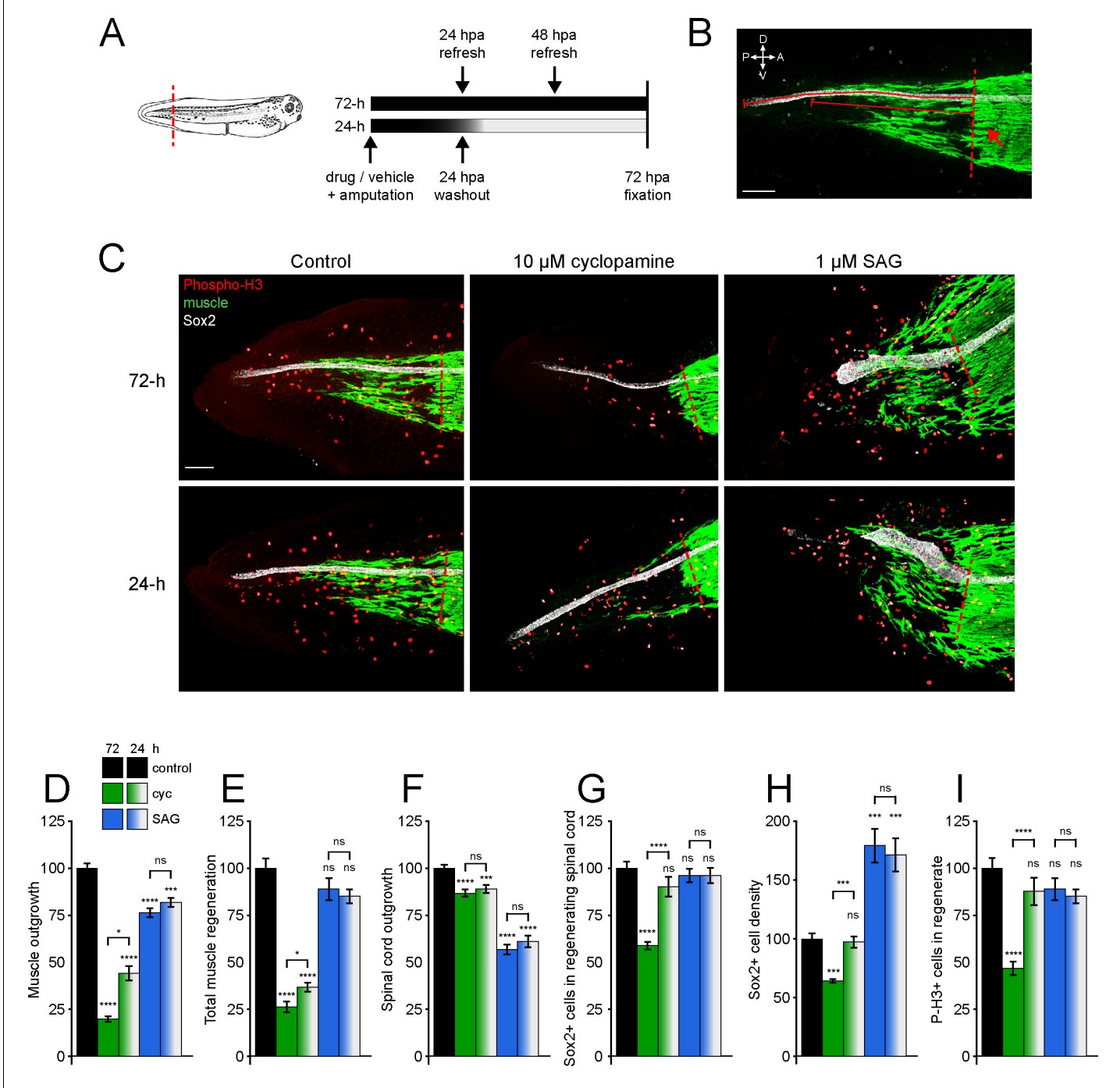

**Figure 1.** Hedgehog signaling regulates spinal cord and muscle regeneration. Stage 39–40 *Xenopus laevis* larvae were incubated for 24 or 72 h after tail amputation in vehicle (0.1% DMSO, Control), antagonist (10 µM cyclopamine, cyc), or agonist (1 µM SAG) of Smoothened (Smo), and immunostained at 72 h post amputation (hpa). (**A**) Schematics of tail amputation and 24 vs. 72 h treatments. (**B**) Measurement of outgrowth for regenerated spinal cord and muscle (solid lines) from amputation plane (dashed line), identified by the posterior-most close-packed band of muscle fibers (arrow). (**C**) Representative z-projections of whole-mount immunostained samples for each group at 72 hpa. Transverse red dashed line indicates amputation plane. Scale bars in (**B, C**), 100 µm. (**D–I**) Graphs show mean ± SEM regenerated muscle outgrowth (**D**) and total new muscle volume (**E**), regenerated spinal cord outgrowth (**F**), total number (**G**), and per length density (**H**) of Sox2+ cells in the regenerated spinal cord, and number of phospho-histone H3+ (P-H3+) cells (**I**) in the regenerate at 72 hpa as % of cohort-matched control. n of larvae: 25–42 per group, N of experiments ≥ 3. *p<0.05, ***p<0.001, ****p<0.0001, ns: not significant, ordinary one-way ANOVA, Brown–Forsythe and Welch ANOVA, or Kruskal–Wallis test, followed by Tukey's, Dunnett's T3, or Dunn's multiple comparisons test, respectively, according to prior normality and equality of SDs tests within and between groups.

*Figure 1 continued on next page*

*Figure 1 continued*

The online version of this article includes the following source data and figure supplement(s) for figure 1:

**Source data 1.** Hedgehog signaling regulates spinal cord and muscle regeneration.
**Figure supplement 1.** Expression of Hedgehog (Hh) ligands is apparent in stage 40 larvae and increases after amputation in the regenerating tail.
**Figure supplement 1—source data 1.** Treatment with Smoothened (Smo) antagonist vismodegib reduces muscle and spinal cord regeneration.
**Figure supplement 2.** Treatment with Smoothened (Smo) antagonist vismodegib reduces muscle and spinal cord regeneration.

Smo modulation alters both muscle and spinal cord regeneration. Inhibition of Smo for 72 hpa with cyclopamine reduces muscle outgrowth into the regenerating tail (*Figure 1D*) and total regenerated muscle volume (*Figure 1E*). Cyclopamine also impairs spinal cord outgrowth (*Figure 1F*) and reduces the total number of NSCs in the newly regenerated spinal cord (*Figure 1G*), resulting in spinal cord with a lower NSC density than in control larvae (*Figure 1H*), which suggests that inhibiting Smo interferes with activation/proliferation of NSCs. Furthermore, treatment of amputated larvae with vismodegib, a structurally distinct Smo antagonist that binds to a different domain than cyclopamine, results in very similar reductions in regeneration (*Figure 1—figure supplement 2*).

In contrast, enhancing Smo activity with SAG has no effect on total muscle regeneration (*Figure 1E*), and even shows a modest inhibitory effect on muscle outgrowth (*Figure 1D*). In addition, SAG treatment universally results in abnormal muscle regeneration consisting in muscle cells that do not follow a longitudinal alignment parallel to the anteroposterior axis and ectopic outgrowth of new muscle fibers dorsally and ventrally from intact axial musculature in the tail stump and regenerated tail, in contrast to all other groups where this phenotype was not observed (26 out of 26 SAG-treated samples; 0 out of 41 controls; 0 out of 31 cyclopamine-treated samples; *Figure 1C*), indicating dysregulated muscle morphogenesis. SAG also reduces the outgrowth of the spinal cord (*Figure 1F*) without altering the total number of new NSCs (*Figure 1G*), resulting in a truncated, wide spinal cord with higher NSC density than in control larvae (*Figure 1C, H*). Similarly, overall mitotic activity at 72 hpa is not affected by SAG but is reduced by cyclopamine (*Figure 1I*). These results argue that Smo activity is essential for NSC proliferation, and that ectopically elevated Smo signaling interferes with the normal progression from NSC proliferation to spinal cord outgrowth. Altogether these results suggest that Smo-mediated signaling is necessary for muscle and spinal cord regeneration.

In addition, we examined the interval during which Hh signaling is necessary for regeneration by comparing 72 h exposure with treatment for only the first 24 h of 72 h total (*Figure 1A*; 24 h vs. 72 h). We found that the effects of 24 h exposure to SAG were comparable to the full 72 h treatment for all metrics of both muscle (*Figure 1D, E*) and spinal cord (*Figure 1F–H*) regeneration, as well as overall cell proliferation in the regenerate (*Figure 1I*), and gave the same abnormal muscle regeneration phenotype with ectopic dorsoventral muscle outgrowth and shortened spinal cord in all samples, in contrast to all other groups where this phenotype was not observed (27 of 27 SAG-treated samples; 0 out of 41 controls; 0 out of 35 cyclopamine-treated samples; *Figure 1C*). Cyclopamine treatment for only 24 hr, however, showed weaker effects than the full 72 hr; the number and density of NSCs in the newly formed spinal cord, as well as the number of mitotic cells in the regenerate, are unchanged by 24 h cyclopamine (*Figure 1G–I*), while reductions in muscle outgrowth and volume are smaller than with 72 h cyclopamine incubation (*Figure 1D, E*). In contrast, the reduction in spinal cord outgrowth is comparable between 24 and 72 h cyclopamine treatments (*Figure 1F*).

These results suggest that endogenous Hh signaling plays a critical role in the early stages of muscle regeneration that involve activation of muscle stem cells, as well as an ongoing role in maintaining the proliferation necessary for both spinal cord and muscle growth, and indicate that balanced Hh signaling is necessary for spinal cord and muscle regeneration.

## Canonical Hh signaling is rapidly repressed following tail amputation

Next, we examine the endogenous activity of canonical Hh signaling following amputation using a dual-cassette transcriptional activity reporter plasmid with simultaneous Gli1/2-dependent expression of enhanced green fluorescent protein (EGFP), and constitutive expression of near-infrared fluorescent protein as a normalizing factor for reporter expression (iRFP670; Gli reporter; *Figure 2A*). When injected into 2–4 cell-stage embryos, this construct is distributed in a mosaic fashion

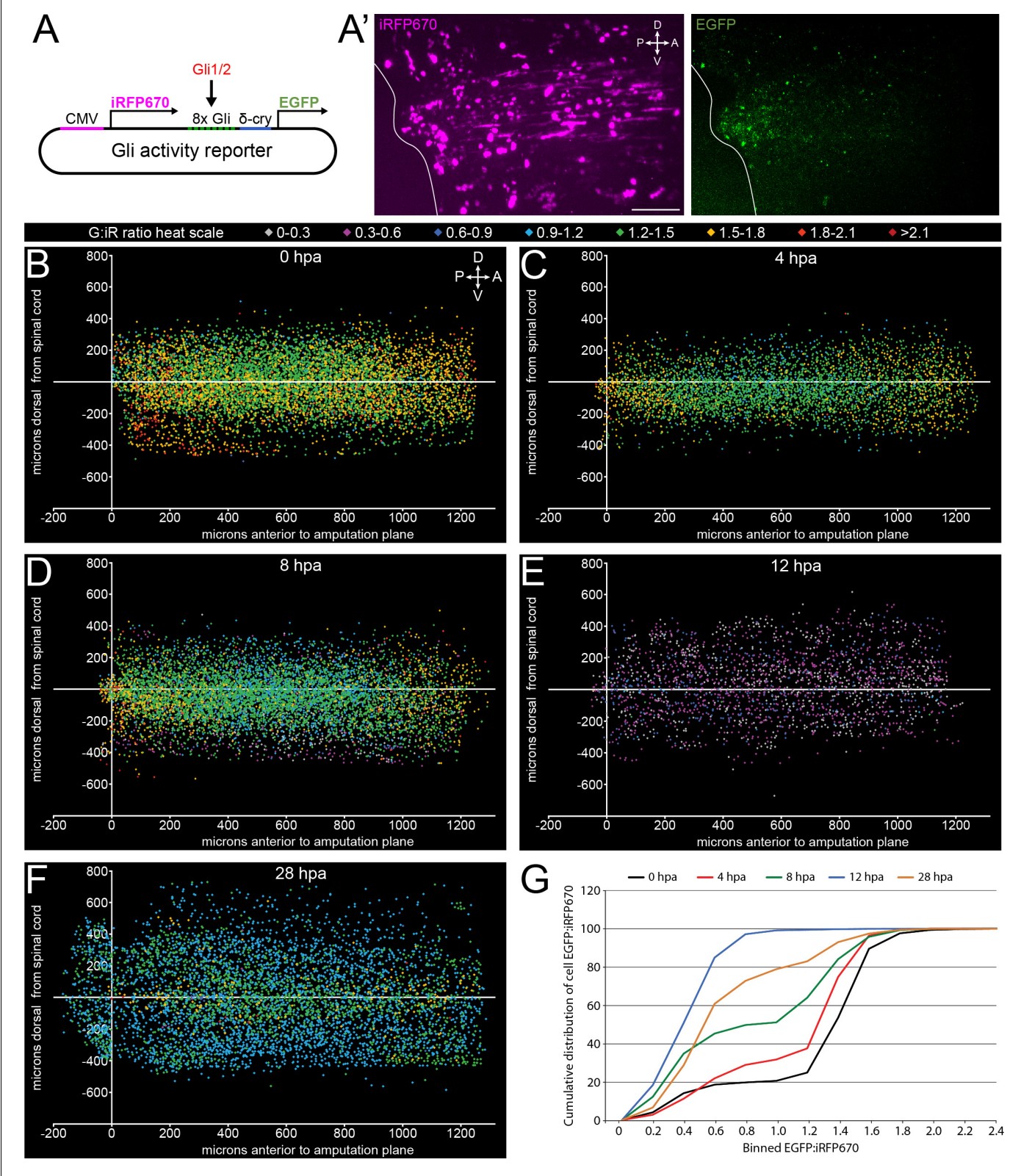

**Figure 2.** Canonical, Gli-dependent Hedgehog signaling is downregulated following tail amputation. Tails of stage 39–40 larvae expressing the Gli transcriptional activity reporter were amputated, and larvae imaged live at intervals from 0 to 28 h post amputation (hpa). (**A**) Schematic of bicistronic reporter plasmid. Constitutive promoter CMV drives expression of iRFP670, and minimal δ-crystallin promoter juxtaposed to eight Gli-binding sites drives expression of enhanced green fluorescent protein (EGFP). (**A'**) Representative z-projections of normalizing factor (iRFP670) and Gli transcriptional

*Figure 2 continued on next page*

*Figure 2 continued*

activity reporter (EGFP) in amputated larval tail at 12 hpa. White outline represents the edge of the tail. Scale bar, 100 μm. (**B–F**) 2D composite of EGFP: iRFP670 (G:iR ratio) intensity displayed by heat scale at indicated hpa from combined iRFP670+ cells. Cell location displayed by position relative to amputation plane (x) and dorsal or ventral to the spinal cord (y). (**G**) Cumulative distribution of EGFP:iRFP670 ratios by time point. All cumulative distribution curves significantly different from control (0 hpa); N of larvae ≥12; p<0.001, all time points compared to 0 hpa, Kolmogorov–Smirnov post-hoc test.

The online version of this article includes the following source data and figure supplement(s) for figure 2:

**Source data 1.** Canonical, Gli-dependent Hedgehog signaling is downregulated following tail amputation.
**Figure supplement 1.** Positive control for Gli transcriptional activity reporter.
**Figure supplement 1—source data 1.** Positive control for Gli transcriptional activity reporter.

throughout the larva, giving an expression-normalized readout of Gli1/2 transcriptional activity (EGFP:iRFP670 ratio, *Figure 2A'*). We validated this construct by treating Gli reporter-injected embryos with SAG, resulting in a significant increase in EGFP:iRFP670 signal during neural plate development, a period known to exhibit high canonical Hh signaling activity (*Lee et al., 1997*; *Belgacem and Borodinsky, 2015*; *Figure 2—figure supplement 1*).

When each iRFP670+ cell is assigned a heat scale color by EGFP:iRFP670 intensity and displayed in two-dimensional space by its position relative to the spinal cord (Y) and the amputation plane (X), we find that relative to 0 hpa (*Figure 2B*), by 4 hpa there is a broad reduction in Gli-reporter signal which appears 300–800 μm anterior to the amputation plane (*Figure 2C*). This decrease expands and deepens through 8 hpa, until reaching a minimum at 12 hpa, before showing partial recovery at 28 hpa (*Figure 2D–F*). When the cumulative distribution of EGFP:iRFP670 ratios is compared by time point, we find significant reductions in the proportion of higher Gli activity cells at all time points post amputation (*Figure 2G*).

These data indicate that amputation rapidly induces widespread inhibition of canonical Gli-dependent signaling.

## Hedgehog-dependent spinal cord and muscle regeneration are primarily non-canonical

Our data suggest that while Smo-dependent Hh signaling is necessary for proper spinal cord and muscle regeneration, canonical, Gli-dependent activity is downregulated immediately following tail amputation; therefore, we directly addressed the necessity of Gli1/2 activity during tail regeneration. GANT61 is a small-molecule inhibitor of the transcriptional activity of Gli1 and Gli2, the primary activators of downstream canonical Hh signaling (*Lauth et al., 2007*). Using our Gli reporter, we demonstrate that treatment with 10 μM GANT61 significantly reduces Gli transcriptional activity in *X. laevis* larvae (*Figure 3—figure supplement 1*).

Treatment of tail-amputated larvae with GANT61 for 72 h (*Figure 3A*) results in only a modest reduction in the number of new NSCs (*Figure 3E*), reducing Sox2 density in the regenerating spinal cord (*Figure 3F*), but does not significantly alter any other regeneration or proliferation metrics in either muscle or spinal cord (*Figure 3B–D, G*). This contrasts sharply with cohort-paired cyclopamine treatment, which shows reductions in all regeneration metrics for muscle and spinal cord (*Figure 3*). Furthermore, when GANT61 treatment is combined with cyclopamine, the results are indistinguishable from cyclopamine treatment alone for all regeneration metrics (*Figure 3*).

In addition to pharmacological inhibition of Gli1/2 activity, we used a translation-blocking morpholino to downregulate protein abundance of Gli2, the primary activator of canonical Hh signaling. As Gli2 signaling is indispensable for early embryogenesis, we coinjected a complementary, UV photolabile blocker morpholino (Photo-MO) which binds to and disables the matched Gli2 morpholino (Gli2-MO), allowing UV-dependent uncaging of Gli2-MO at later developmental periods (*Figure 4A*). Exposure of uninjected embryos to UV does not reduce any regeneration metrics (*Figure 4—figure supplement 1A–D*), while activation of Gli2-MO has no significant effect on regeneration of muscle or spinal cord, with the exception of a small reduction in spinal cord outgrowth (*Figure 4C–H*), despite downregulation of full-length Gli2 protein levels (*Figure 4—figure supplement 1E*).

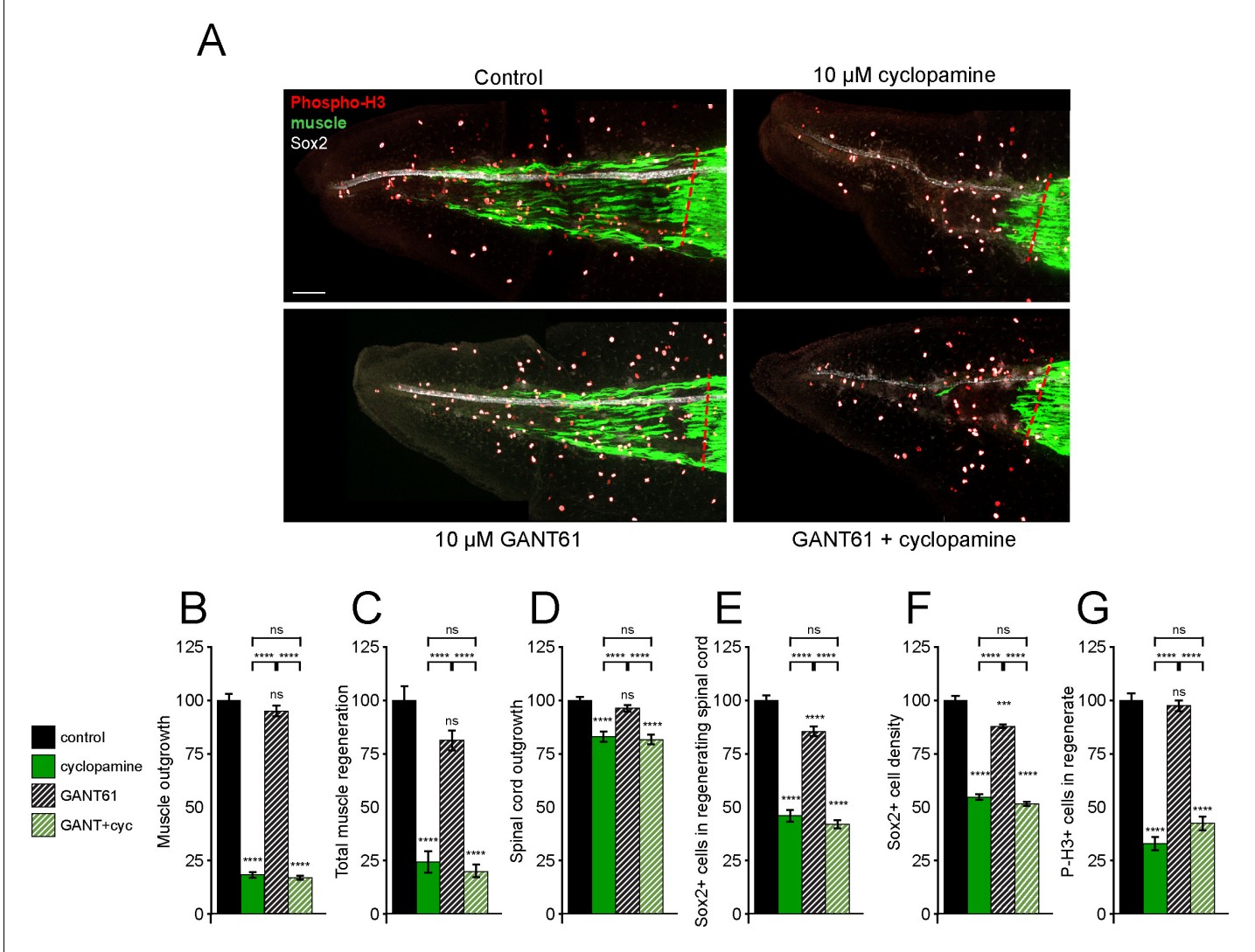

**Figure 3.** Gli1/2 transcriptional activity is not necessary for spinal cord and muscle regeneration. Stage 39–40 larvae were incubated for 72 h after tail amputation in vehicle (0.1% DMSO, Control) or Gli1/2 antagonist (10 μM GANT61, GANT) and/or 10 μM cyclopamine (cyc), then whole-mount immunostained. (**A**) Images show representative z-projections for each group at 72 h post amputation (hpa). Transverse red dashed line indicates amputation plane. Scale bar, 100 μm. (**B–G**) Graphs show mean ± SEM regenerated muscle outgrowth (**B**) and total volume (**C**), regenerated spinal cord outgrowth (**D**), total number (**E**) and per length density (**F**) of Sox2+ cells in the regenerated spinal cord, and overall number of phospho-histone H3 + (P-H3+) cells (**G**) in the regenerate at 72 hpa as % of cohort-matched control, n of larvae: 13–31 per group, N of experiments ≥ 3. ***p<0.001, ****p<0.0001, ns: not significant, ordinary one-way ANOVA, Brown–Forsythe and Welch ANOVA, or Kruskal–Wallis test, followed by Tukey's, Dunnett's T3, or Dunn's multiple comparisons test, respectively, according to prior normality and equality of SDs within and between groups.

The online version of this article includes the following source data and figure supplement(s) for figure 3:

**Source data 1.** Gli1/2 transcriptional activity is not necessary for spinal cord and muscle regeneration.
**Figure supplement 1.** Treatment with GANT61 inhibits Gli1/2 transcriptional activity.
**Figure supplement 1—source data 1.** Treatment with GANT61 inhibits Gli1/2 transcriptional activity.

These findings indicate that canonical Hh signaling is, for the most part, not necessary for spinal cord and muscle regeneration, and suggest that Smo-dependent regeneration recruits a non-canonical signaling pathway.

## PKA signaling is necessary for regeneration

In previous studies, we found that $Ca^{2+}$ activity is associated with both muscle regeneration (***Tu and Borodinsky, 2014***) and PKA-dependent non-canonical Shh signaling through Smo during embryonic

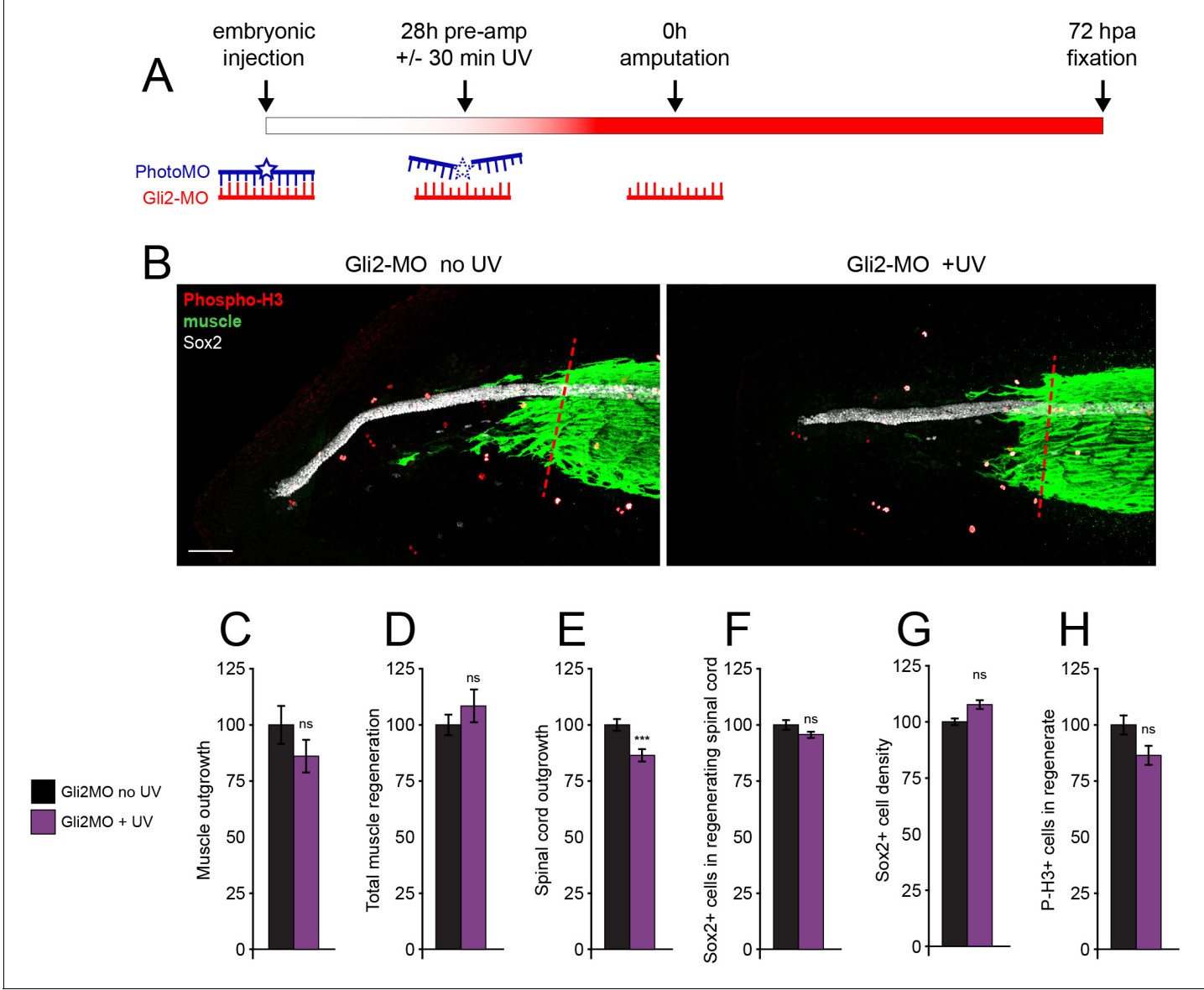

**Figure 4.** Genetic downregulation of canonical Hedgehog-Smoothened (Hh-Smo) signaling does not affect spinal cord and muscle regeneration. Larvae containing Gli2 morpholino (Gli2-MO) bound to photo-morpholino were UV illuminated (+UV) or not (control, no UV) for 30 min, 28 h before (stage 28) amputation (stage 39–40), to uncage morpholino and induce downregulation of Gli2 expression. (**A**) Schematic of time course of morpholino activation and amputation. (**B**) Images show representative z-projections of samples for each group at 72 h post amputation (hpa). Transverse red dashed line indicates amputation plane. Scale bar, 100 μm. (**C–H**) Graphs show mean ± SEM regenerated muscle outgrowth (**C**) and total volume (**D**), regenerated spinal cord outgrowth (**E**), total number (**F**) and per length density (**G**) of Sox2+ cells in the regenerated spinal cord, and overall number of phospho-histone H3+ (P-H3+) cells (**H**) in the regenerate at 72 hpa as % of cohort-matched control, n of larvae: 16–35 per group, N of experiments ≥ 3. ***$p<0.001$, ns: not significant, unpaired t-test, Welch's t-test, or Kolmogorov–Smirnov test, according to prior normality and equality of SDs tests within and between groups.

The online version of this article includes the following source data and figure supplement(s) for figure 4:

**Source data 1.** Genetic downregulation of canonicalHedgehog-Smoothened(Hh-Smo)signaling does not affect spinal cord and muscle regeneration.
**Figure supplement 1.** UV treatment does not reduce regeneration and uncages Gli2-MO, knocking down Gli2 expression.
**Figure supplement 1—source data 1.** UV treatment does not reduce regeneration and uncages Gli2-MO, knocking down Gli2 expression.

spinal cord development (*Belgacem and Borodinsky, 2011*; *Belgacem and Borodinsky, 2015*). To examine a potential role for PKA activity during tail regeneration, we treated amputated larvae with the PKA inhibitor KT5720 (*Figure 5A*). Inhibiting PKA diminishes all regeneration metrics, reducing outgrowth of muscle (*Figure 5B*) and spinal cord (*Figure 5D*), total muscle regeneration (*Figure 5C*), and spinal cord NSC (*Figure 5E*) and mitotic cell (*Figure 5G*) counts. The reduction in NSC number when PKA is inhibited is approximately proportional to the reduction in spinal cord outgrowth, resulting in a slight increase in NSC density (*Figure 5F*), likely due to the fact that PKA inhibition leads to a larger overall decrease in spinal cord regeneration than when inhibiting Smo (*Figure 5D*).

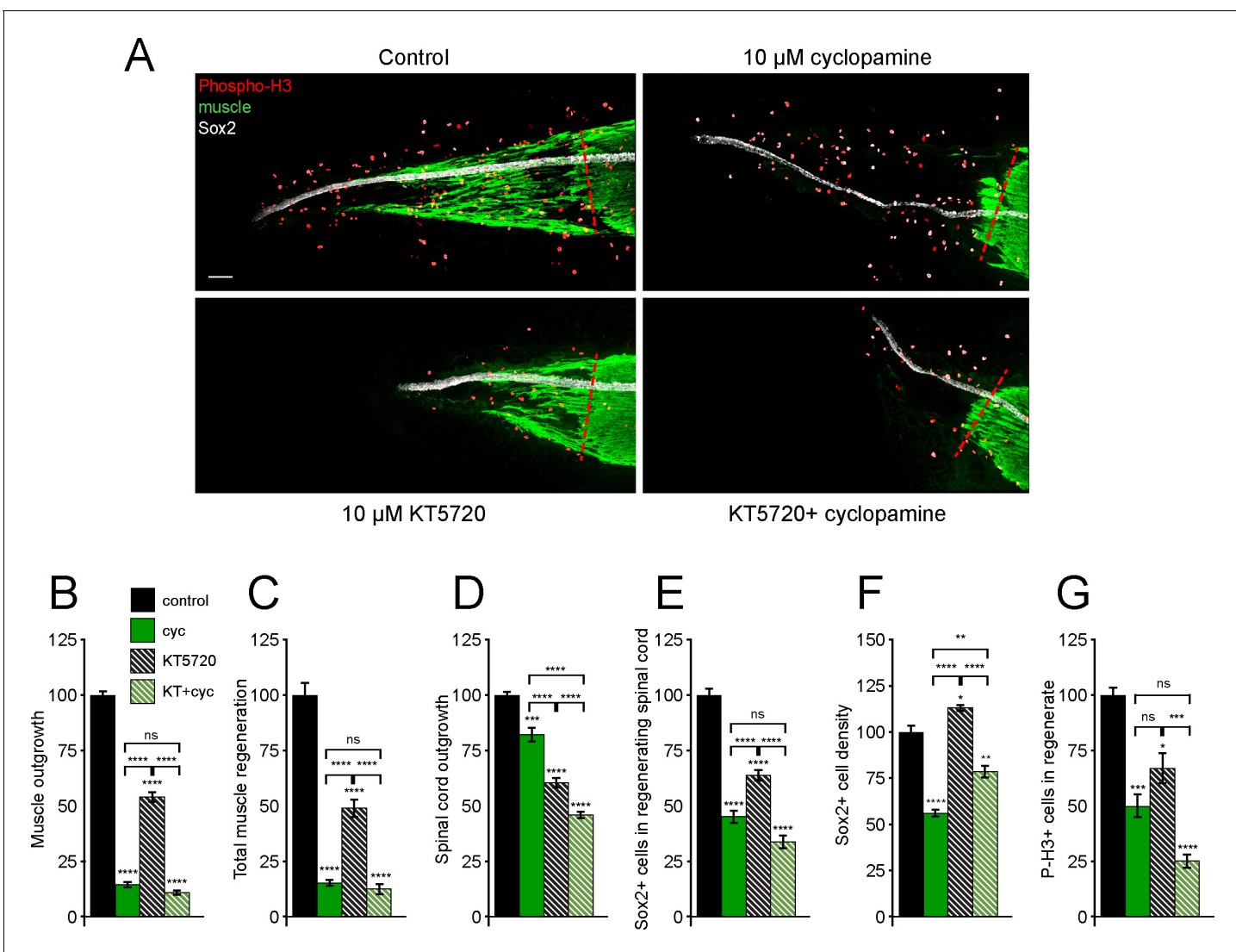

**Figure 5.** Tissue-specific interaction between PKA and Smoothened (Smo) signaling in regulation of spinal cord and muscle regeneration. Stage 39–40 larvae were incubated for 72 h after tail amputation in vehicle (0.1% DMSO, Control), PKA antagonist (10 µM KT5720, KT), or/and 10 µM cyclopamine (cyc). (A) Images show representative samples for each group at 72 h post amputation (hpa). Transverse red dashed line indicates amputation plane. Scale bar, 100 µm. (B–G) Graphs show mean ± SEM regenerated muscle outgrowth (B) and total volume (C), regenerated spinal cord outgrowth (D), total number (E) and per length density (F) of Sox2+ cells in the regenerated spinal cord, and overall number of phospho-histone H3+ (P-H3+) cells (G) in the regenerate at 72 hpa as % of cohort-matched control, n of larvae: 11–29 per group, N of experiments ≥ 3. *p<0.05, **p<0.01, ***p<0.001, ****p<0.0001, ns: not significant, ordinary one-way ANOVA, Brown–Forsythe and Welch ANOVA, or Kruskal–Wallis test, followed by Tukey's, Dunnett's T3, or Dunn's multiple comparisons test, respectively, according to prior normality and equality of SDs within and between groups.
The online version of this article includes the following source data for figure 5:

**Source data 1.** Tissue-specific interaction between PKA and Smoothened(Smo)signaling in regulation of spinal cord and muscle regeneration.

This suggests that PKA participates in spinal cord regeneration beyond the initial activation and proliferation of NSCs, while Smo signaling is particularly important for NSC proliferation.

Simultaneous treatment with KT5720 and cyclopamine reveals tissue-specific epistasis: inhibition of Smo is epistatic over PKA inhibition for muscle outgrowth (*Figure 5B*) and total new muscle formation (*Figure 5C*), suggesting that PKA acts upon muscle regeneration downstream of Smo. In contrast, dual KT5720/cyclopamine treatment results in additive inhibition of spinal cord outgrowth (*Figure 5D*), but no significant additivity in reduction of NSC (*Figure 5E*) or mitotic cell counts (*Figure 5G*). Interestingly, simultaneous inhibition of PKA and Smo slightly ameliorates the reduction in NSC density observed with cyclopamine alone (*Figure 5F*), reinforcing the concept that PKA acts upon spinal cord regeneration in a mechanistically distinct manner from Smo. This suggests that PKA and Smo act on spinal cord regeneration in a parallel but coordinated manner. It should be noted that KT75720 has been shown to also inhibit other kinases in cell line and cell-free studies, including phosphorylase kinase, PDK1, and MEK (*Davies et al., 2000*; *Murray, 2008*). Hence, further investigation will be necessary to sort out the participation of specific alternative kinase pathways in tissue regeneration.

Since kinases such as PKA are known to be potent regulators of a wide variety of signaling pathways, we directly examined one of PKA's primary downstream effectors, the transcription factor CREB, which is activated by non-canonical Hh signaling in the embryonic spinal cord (*Belgacem and Borodinsky, 2015*). To examine endogenous CREB activity following amputation, we fixed larvae pre-amputation (0 hr) and at intervals from 4 to 48 hpa and stained for activated phospho-Ser133-CREB (P-CREB; *Belgacem and Borodinsky, 2015*; *Gonzalez and Montminy, 1989*). We found that P-CREB signal is strong in the skin around the regenerating tail tip, making isolation of P-CREB signal in muscle unfeasible (*Figure 6—figure supplement 1*). However, we were able to isolate P-CREB staining from the Sox2-labeled region of the spinal cord (*Figure 6A*). We find that the density of P-CREB+ cells in the spinal cord within the amputated tail stump, 100 μm or more anterior from the amputation plane, is similar across individual 100 μm bins of spinal cord in all time points post amputation, comparable to pre-amputation controls (*Figure 6B*). In contrast, within the amputation region and regenerating spinal cord, we observe a significant increase in P-CREB+ cells compared to the intact stump as early as 4 hpa and throughout the first 24 hpa (*Figure 6B*). By 48 hpa, this increase in P-CREB+ cell density is no longer apparent at any region in the regenerating spinal cord (*Figure 6B*). This suggests that CREB activity is differentially regulated spatially and temporally in the spinal cord, with a rapid, transient increase in CREB activation in the spinal cord proximal to the amputation site.

Moreover, phosphorylation of P-CREB immediately post-amputation is bidirectionally dependent on Smo activity, as shown in transverse sections of paraffin-embedded larvae treated with SAG or cyclopamine for 4 hpa. Within 400 μm of the amputation site, cyclopamine-treated larvae show significantly fewer P-CREB+ cells in the spinal cord relative to vehicle controls, while SAG treatment increases this metric (*Figure 7A, B*). Analysis of 24 hpa whole mounts reveals a similar effect, with cyclopamine treatment reducing P-CREB+ cell counts in regenerated spinal cord, while treatment with SAG increases CREB activation compared to controls (*Figure 7C, D*). These data indicate that Hh signaling regulates CREB activation during regeneration.

To further examine the potential for non-canonical Hh signaling-dependent interplay between regeneration of spinal cord and muscle, we compared the correlation in the regeneration of these tissues when either Smo or PKA signaling has been altered. Under control conditions, we observe a close correlation between muscle and spinal cord outgrowth when examined on a per-sample basis. This correlation is decoupled by enhancing Smo signaling, which allows muscle outgrowth in the absence of normal spinal cord outgrowth, and by inhibiting it, which blocks muscle regeneration almost entirely, despite substantial spinal cord outgrowth (*Figure 8*). In contrast, PKA inhibition reduces all regeneration metrics without altering the interrelationship between the extent of regeneration of both tissues. Inhibiting Gli1/2 also has no discernible effect on correlated outgrowth between the two tissues (*Figure 8*). This suggests that balanced non-canonical Smo-dependent Hh signaling is necessary for coordinated regeneration of spinal cord and muscle (*Figure 9*).

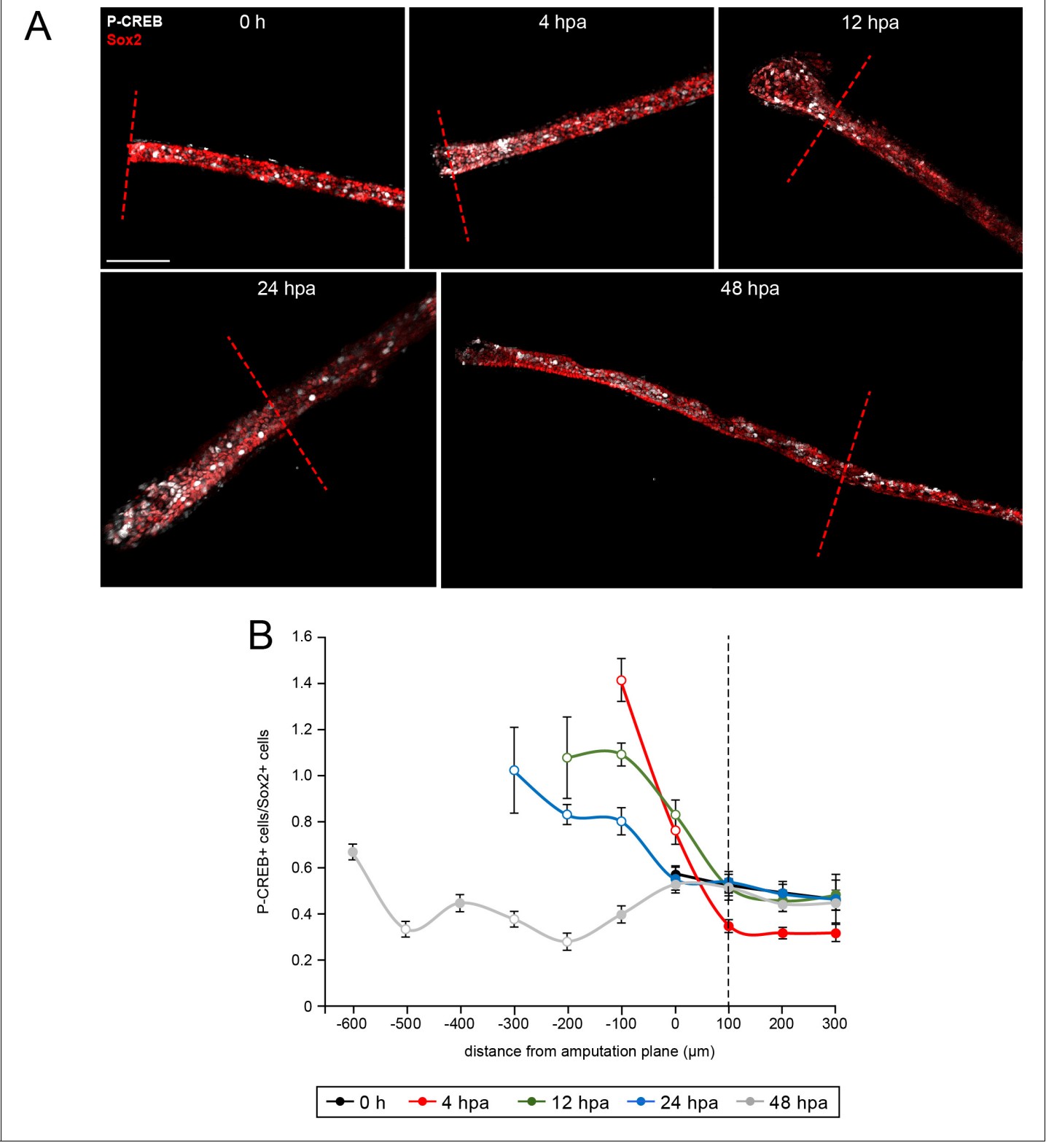

**Figure 6.** Spatiotemporal activation of CREB in the injured and regenerated spinal cord. Stage 39–40 larvae were amputated and processed for whole-mount immunostaining at the indicated hour post amputation (hpa), with the exception of the 0 h group, which was first fixed and then amputated to represent the pre-amputation group. (**A**) Images show representative z-projections of immunostained samples for each group that were digitally processed to isolate spinal-cord associated P-CREB. Transverse red dashed line indicates amputation plane. Scale bar, 100 μm. (**B**) Graph shows mean ± SEM number of total P-CREB+ cells normalized to number of Sox2+ cells in 100 μm sections of spinal cord anterior (positive) and posterior

*Figure 6 continued on next page*

*Figure 6 continued*

(negative) to the amputation plane (0). N of larvae ≥5 per group. Open circles denote p<0.05 vs. time-matched average number of P-CREB+/Sox2 + cells within the region 100–200 µm anterior (100 µm, dashed black line) to the amputation plane (0 µm), one-way ANOVA.

The online version of this article includes the following source data and figure supplement(s) for figure 6:

**Source data 1.** Spatiotemporal activation of CREB in the injured and regenerated spinal cord.
**Figure supplement 1.** Unedited images from samples featured in *Figure 6*.

## Discussion

Our work demonstrates that Hh signaling is necessary for regeneration of muscle and spinal cord in *Xenopus* larvae. These findings are in keeping with previous discoveries showing Hh signaling involvement in the regeneration of a variety of tissues including liver, heart, and limb (*Grzelak et al., 2014*; *Singh et al., 2012*; *Singh et al., 2018b*), in addition to *Xenopus* larval tail (*Taniguchi et al., 2008*; *Taniguchi et al., 2014*), as well as the fact that Hh pathway activation is enhanced at the site of tail amputation in *X. laevis* (*Taniguchi et al., 2014*) and zebrafish (*Romero et al., 2018*). Recent studies have also shown that Smo and Ptch1 experience a transient increase in transcriptional availability immediately after tail amputation in *Xenopus tropicalis* (*Kakebeen et al., 2020*), and shh transcript level increases in 1 dpa *X. laevis* larvae (*Aztekin et al., 2019*), further emphasizing the importance of this pathway during regeneration.

The regenerated spinal cord under Smo inhibition exhibits around half of the number of NSCs at 72 hpa compared to control animals. This suggests that with regards to spinal cord regeneration Hh-Smo signaling is particularly important for NSC proliferation. Moreover, pharmacological enhancement of Smo signaling leads to simultaneous NSC proliferation and blockade of spinal cord morphogenesis, further supporting the model of Smo signaling acting predominantly on NSC activation and proliferation upon injury (*Figure 9*).

Smo signaling is also essential for muscle regeneration, and arguably for the initial stages of muscle stem cell activation and proliferation since inhibiting Smo signaling during the first 24 hpa is enough to strongly decrease the replenishing of skeletal muscle (*Figure 9*). Moreover, it appears that the first 24 h post injury are a critical period for Smo-dependent muscle stem cell activation, unlike NSCs, which seem to be able to be activated and proliferate even after an initial 24 h of Smo inhibition. It remains to be determined whether Smo-dependent muscle regeneration is acting directly on muscle stem cells or indirectly through signals from the spinal cord. Nevertheless, the evidence that in the presence of overactive Smo both the spinal cord and muscle exhibit aberrant outgrowth, along with published work from others demonstrating the necessity of the spinal cord for proper tail regeneration (*Taniguchi et al., 2008*), supports a potential interaction between Smo-mediated spinal cord and muscle regeneration (*Figure 9*).

Additionally, we demonstrate that canonical, Gli-dependent Hh signaling is endogenously down-regulated after tail amputation, and further inhibiting this pathway either pharmacologically or genetically only marginally affects muscle and spinal cord regeneration, suggesting that Hh-Smo is acting primarily through a Gli-independent pathway (*Figure 9*). In contrast, CREB activity is recruited immediately after injury, at the amputation site and within the regenerating spinal cord, and fades within the first 24 h of spinal cord regeneration. Moreover, PKA activity, unlike Gli, is necessary for efficient regeneration of both muscle and spinal cord, and CREB is activated upon injury in amputation-proximal and regenerating neural tissue in a Smo-dependent manner. This distinct recruitment of a non-canonical Hh signaling pathway resembles the switch in Hh signaling observed during embryonic stages of early spinal cord development, when canonical, Gli-activating Shh signaling is restricted to the early neural plate (*Balaskas et al., 2012*; *Belgacem and Borodinsky, 2015*; *Lee et al., 1997*), but is later repressed through a Shh-$Ca^{2+}$-PKA-CREB signaling axis that mediates Shh-dependent spinal cord neuron differentiation (*Belgacem and Borodinsky, 2011*; *Belgacem and Borodinsky, 2015*).

PKA recruitment is not the only potential mediator of Hh non-canonical, Smo-dependent regeneration. The effects of inhibiting PKA on muscle regeneration are lower in magnitude compared with Smo inhibition, with the latter resulting in almost complete blockade of muscle replenishment in the regenerated tail. This may be the reason for apparent epistasis when simultaneously inhibiting Smo and PKA pathways in regulating muscle regeneration. Alternatively, PKA may be one of multiple

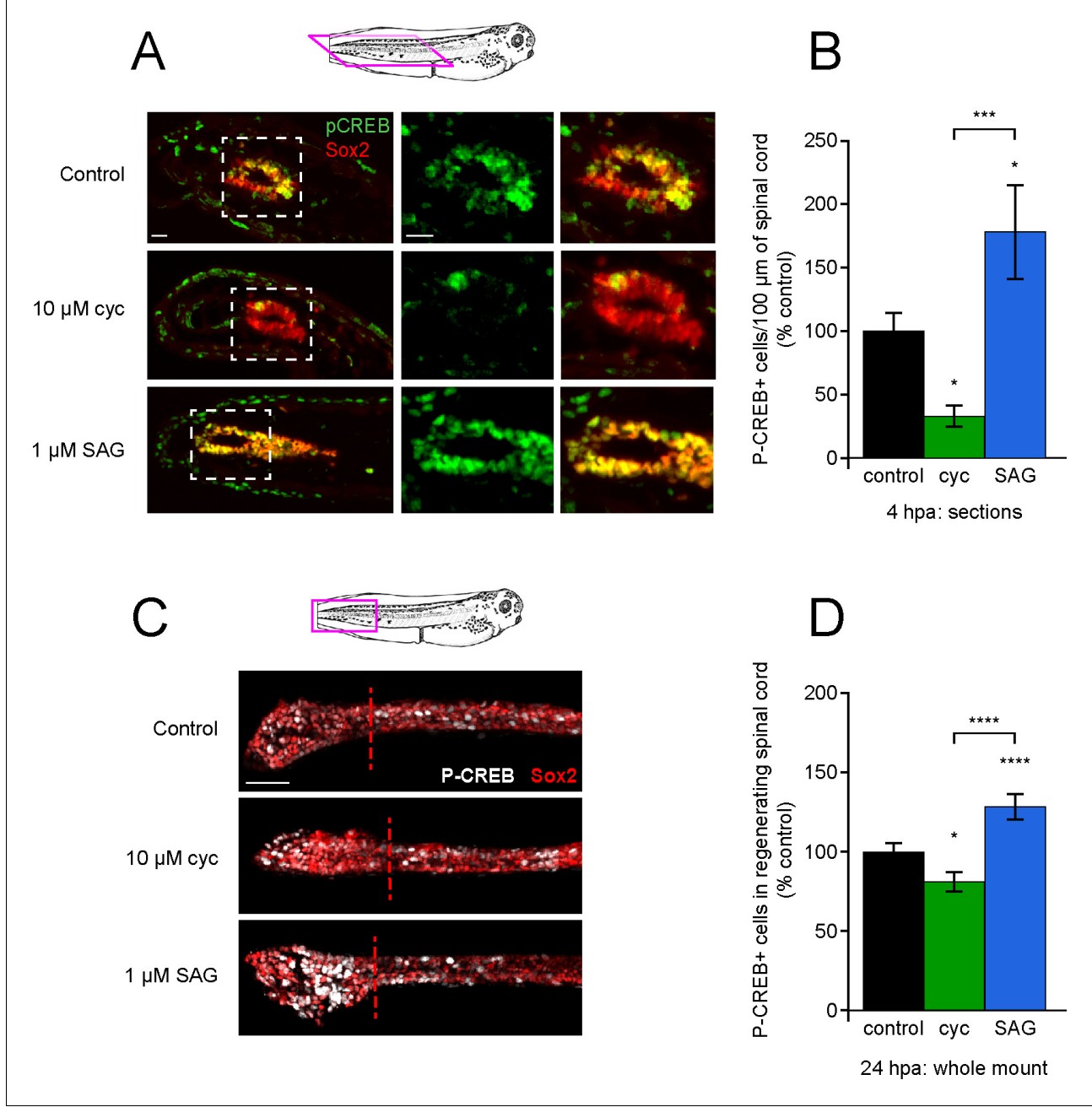

**Figure 7.** Post-amputation activation of CREB is regulated by Hedgehog (Hh) signaling. Stage 39–40 larvae were amputated in vehicle (0.1% DMSO, control), 10 μM cyclopamine, or 1 μM SAG, then fixed at either 4 or 24 h post amputation (hpa) and processed for immunostaining in either paraffin sections (A, B), or whole-mounts (C, D), respectively. (A) Representative longitudinal sections from 4 hpa. Sample orientation displayed in schematic. Scale bars, 20 μm. (B) Graph shows mean ± SEM P-CREB+ cell count in 100-μm-long spinal cord within 400 μm of the amputation plane as % of control, n of larvae ≥5 per group from N = 3 independent experiments. (C) Representative whole-mount sections from 24 hpa. Sample orientation displayed in schematic. Scale bar, 50 μm. (D) Graph shows mean ± SEM P-CREB+ cell count in the regenerated spinal cord, normalized to cohort matched controls, n of larvae ≥12 per group from N = 3 separate experiments. In (B, D), *p<0.05, ***p<0.001, ****p<0.0001, one-way ANOVA + Holm–Sidak's multiple comparison.

The online version of this article includes the following source data for figure 7:

**Source data 1.** Post-amputation activation of CREB is regulated by Hedgehog(Hh)signaling upon injury.

**Source data 2.** Post-amputation activation of CREB is regulated by Hedgehog(Hh)signaling in the regenerating spinal cord.

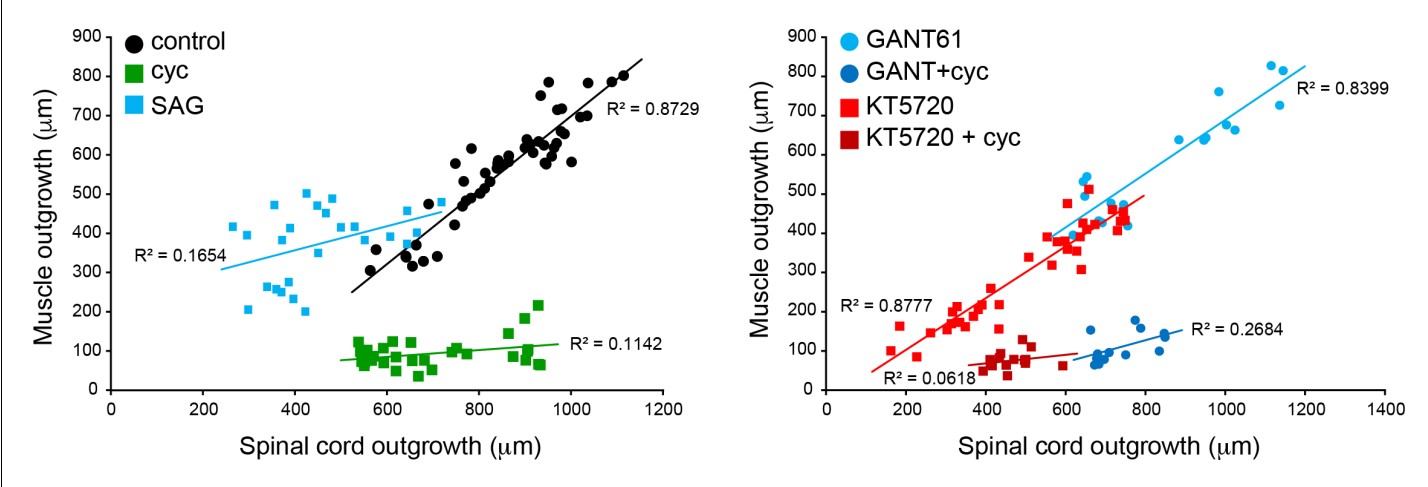

**Figure 8.** Coordination of spinal cord and muscle outgrowth in the regenerating tail is dependent on Smoothened (Smo) signaling. Individual samples for each treatment are displayed by total outgrowth for spinal cord (X) and muscle (Y). Simple linear regression lines are fit for each treatment, and $R^2$ goodness-of-fit values displayed by each line. N = 13–39 samples per group.

The online version of this article includes the following source data for figure 8:

**Source data 1.** Coordination of spinal cord and muscle outgrowth in the regenerating tail is dependent on Smoothened(Smo)signaling.

non-canonical Hh signaling pathways downstream of Smo acting on muscle regeneration (*Figure 9*). It should also be noted that KT5720 has been demonstrated to effectively inhibit other kinases in purified, cell-free preparations (*Davies et al., 2000*); among these is GSK3β, which is known to regulate Hh signaling through both Gli2 (*Riobó et al., 2006*) and its inhibitor SUFU (*Chen et al., 2011*). The involvement of non-canonical Hh signaling and CREB supports the involvement of PKA (*Belgacem and Borodinsky, 2015*), but further experiments would be necessary to rule out or include contributions by other kinases.

The link between Smo and regeneration must lie in downstream transcription factors other than Gli, such as NFKB (*Qu et al., 2013*), MycN (*Mani et al., 2020*; *Singh et al., 2018b*), CREB (*Belgacem and Borodinsky, 2015*), or JAK/STAT (*Tian et al., 2015*), which has previously been shown to regulate regeneration in *X. laevis* (*Tapia et al., 2017*). If specific signaling pathways downstream of Hh-dependent regeneration can be targeted, it may help limit the pleiotropic effects observed with direct manipulation of Smo (*Wu et al., 2017*).

Finally, PKA activity in spinal cord regeneration does not appear to be linearly downstream to Smo-dependent signaling (*Figure 9*) since we observed additive effects when simultaneously inhibiting Smo and PKA. The exact mechanisms by which PKA modulates regeneration, however, remain unclear. PKA may act as an inhibitor of Gli, which would be in keeping with both our data showing Gli downregulation following amputation and its developmental role as a repressor of canonical Hh signaling (*Hammerschmidt et al., 1996*), as well as PKA participation in the non-canonical, $Ca^{2+}$ activity and CREB-dependent repression of Gli during *X. laevis* spinal cord development (*Belgacem and Borodinsky, 2015*; *Figure 9*). Since we have also found that $Ca^{2+}$ activity is necessary for *Xenopus larva* tail regeneration following amputation (*Tu and Borodinsky, 2014*), our results showing an early reduction in Gli activity and activation of CREB, as well as the dependence of spinal cord and muscle regeneration on PKA activity, suggest that the aforementioned non-canonical pathway may be at least partially responsible for tissue regeneration.

Manipulation of Hh signaling has already shown great promise in treating a variety of conditions, including cancer (*Ruat et al., 2014*), neural injury and stroke (*Bambakidis and Onwuzulike, 2012*), cardiac ischemia (*Dunaeva and Waltenberger, 2017*), appendage regeneration (*Chen et al., 2014*; *Singh et al., 2015*), and even osteoporosis and obesity (*Hadden, 2014*). Based on the results from this study, we predict that selectively enhancing regeneration-specific, non-canonical Hh signaling in

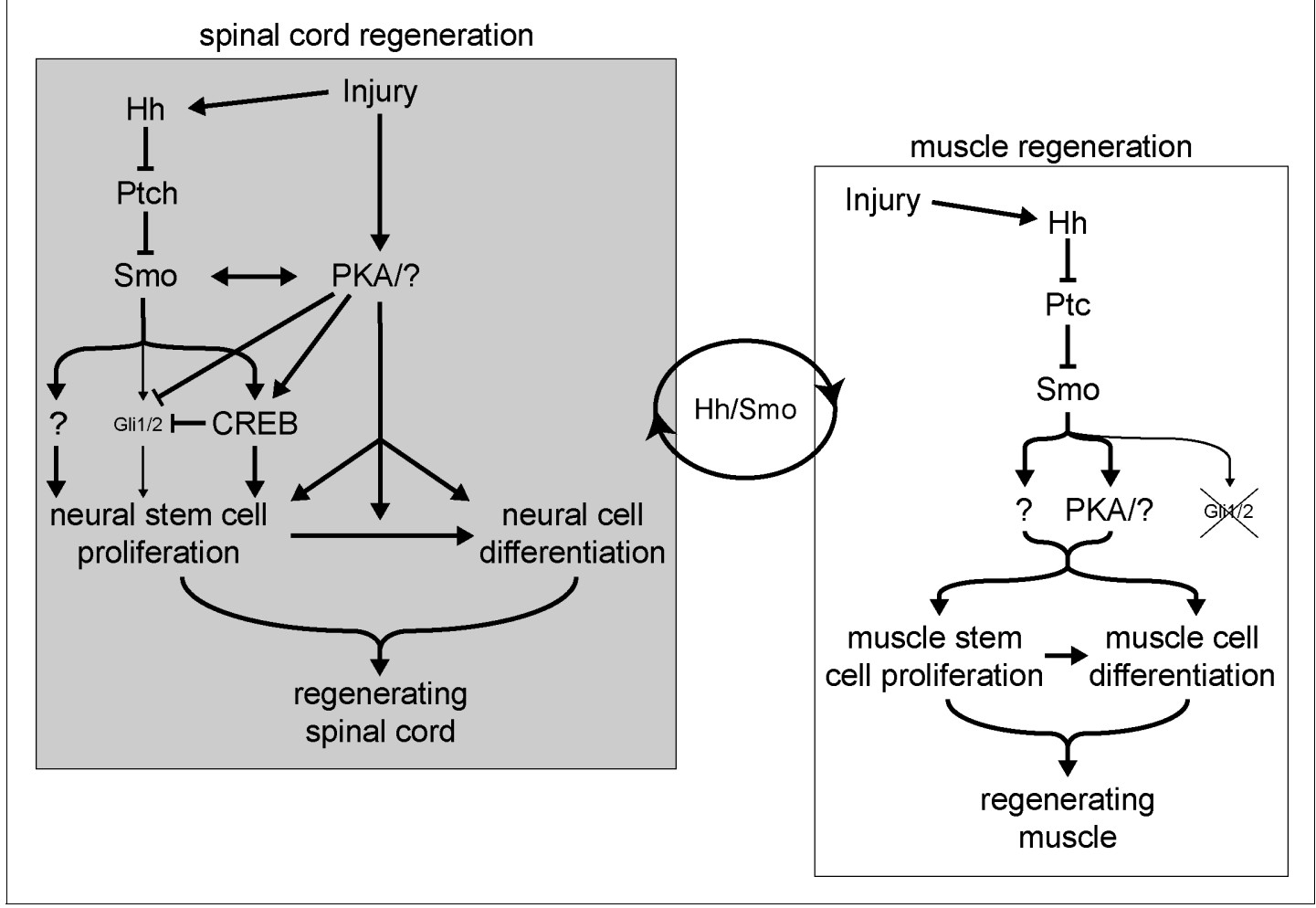

**Figure 9.** Model for Hedgehog (Hh)-dependent regulation of spinal cord and muscle regeneration. Injury recruits non-canonical Hh signaling in the spinal cord to activate neural stem cells for replenishing the regenerating spinal cord. Injury also activates PKA which acts independently from Smoothened (Smo) on neural stem cell proliferation to promote spinal cord regeneration. Non-canonical Hh signaling is also essential for muscle regeneration starting from the initial stages post injury, where PKA appears to be downstream of Smo activation. The coordination between the magnitude of regenerating spinal cord and muscle outgrowth is dependent on non-canonical Hh/Smo signaling.

spinal cord and muscle during a critical period following injury might promote the repair and replenishment of functional tissues.

## Materials and methods

**Key resources table**

| Reagent type (species) or resource | Designation | Source or reference | Identifiers | Additional information |
|---|---|---|---|---|
| Antibody | 5E1 (mouse monoclonal) | Developmental Studies Hybridoma Bank | RRID:AB_528466 | Shh; (1:50) |
| Antibody | GAPDH (goat polyclonal) | SICGEN | RRID:AB_0049-200 | Loading control whole-cell lysates, western blots (1:1000) |

*Continued on next page*

*Continued*

| Reagent type (species) or resource | Designation | Source or reference | Identifiers | Additional information |
|---|---|---|---|---|
| Antibody | 12/101 (mouse monoclonal) | Developmental Studies Hybridoma Bank | RRID:AB_531892 | Skeletal muscle; (1:100) |
| Antibody | Sox2 (goat polyclonal) | R&D | AF2018, RRID:AB_355110 | Neural stem cells; (1:300-1:400) |
| Antibody | P-H3 (rabbit polyclonal) | Millipore | 06-570, RRID:AB_310177 | Mitotic marker; (1:400) |
| Antibody | P-CREB (rabbit polyclonal) | Cell Signaling | 9198 | Phosphorylated transcription factor; (1:800-1:1500) |
| Antibody | Gli2 (goat polyclonal) | R&D | AF3635; RRID:AB_211902 | Transcription factor; (1:800) |
| Antibody | Lamin-B1 (rabbit monoclonal) | Cell Signaling | 9087; RRID:AB_10896336 | Nuclear protein for loading control in western blot assays; (1:500) |
| Recombinant DNA reagent | p8xGli-EGFP_CMV-iRFP670 | This paper | | Gli activity reporter; design described in Materials and methods/ Gli activity reporter section of this paper |
| Sequence-based reagent | Gli2-morpholino | Gene Tools | | GCACAGAACGCAGGTAATGCTCCAT |
| Sequence-based reagent | Gli2-photo-morpholino | Gene Tools | | ATGGAGCATTACPTGCGTTCT |
| Chemical compound, drug | Cyclopamine | Sigma-Aldrich | C4116 | 10–20 mM stock in DMSO |
| Chemical compound, drug | SAG | Calbiochem | 566660 | 5 mM stock in $H_2O$ |
| Chemical compound, drug | Vismodegib | Sigma | 879085-55-9 | 50 mM in DMSO |
| Chemical compound, drug | KT5720 | Tocris | 1288 | 10 mM stock in DMSO |
| Chemical compound, drug | GANT61 | Tocris | 3191 | 10 mM srock in DMSO |
| Chemical compound, drug | Tricaine-S | Syndel | MS 222 | Anesthetic |

## Animals

*X. laevis* females were primed (50 units) and injected (350–400 units) with human chorionic gonado-tropin to induce egg laying. Eggs were squeezed into 1× Marc's Modified Ringer solution (MMR in mM: 110 NaCl, 2 KCl, 1 $MgSO_4$, 2 $CaCl_2$, 5 HEPES, 0.1 EDTA; pH to 7.8), fertilized with minced testis, and allowed to develop at a controlled temperature in 10% MMR. Mixed sex specimens were used at either 12–18 h (Nieuwkoop-Faber [NF] stage 11–16 embryos) or 60 h (NF stage 39–40 larvae) post fertilization.

### Tail amputation and pharmacological treatments

NF stage 39–40 larvae were anesthetized with 0.02% tricaine methanesulfonate (TMS, Syndel) with or without the specified drugs until non-responsive. Larvae were then amputated under a dissection stereoscope using a scalpel blade at approximately 1/4 of the length from the tail tip, where the tail begins to taper. The anesthetic was then washed out and the amputated larvae were incubated at 21–23°C in 10% MMR with vehicle or 10 µM cyclopamine (from 10 to 20 mM stock in DMSO; Sigma C4116), 20 µM vismodegib (from 50 mM stock in DMSO; Sigma 879085-55-9), 1 µM SAG (from 5 mM stock in $H_2O$; Calbiochem 566660), 10 µM GANT61 (from 10 mM stock in DMSO; Tocris 3191), and/or 10 µM KT5720 (from 10 mM stock in DMSO; Tocris 1288). For multi-day treatments, solutions were replaced daily.

### Regeneration analysis

The amputation plane was defined by the most posterior region of close-packed, organized bands of 12/101 stained muscle cells as regenerated muscle lacks this structure. Outgrowth was defined as the distance between the amputation plane and either the furthest posterior Sox2+ cells associated with the regenerating spinal cord (spinal cord outgrowth) or the furthest posterior 12/101+ muscle cells (muscle outgrowth). Total muscle regeneration was defined as the volume of muscle cells in the regenerate posterior to the amputation plane, not including dorsal/ventral outgrowth of muscle from the intact chevrons in SAG treatment conditions. Sox2 and P-H3 counts were performed respectively on spinal cord and total regenerate posterior to the amputation plane. All these measurements were done using the image analysis software Imaris.

### Whole-mount immunostaining

Larvae were anesthetized in 0.02% TMS in 10% MMR, then fixed in 3.7% formaldehyde in 1× MEMFA saline (100 mM MOPS, 2 mM EGTA, 100 mM $MgSO_4$) overnight at 4°C. Samples were bleached overnight in $H_2O_2$/Dent's fixative, then permeabilized with 0.5% Triton X100 in 1× PBS (PBT) and blocked in 0.5% PBT + 2% BSA. Primary and secondary antibody incubations were performed in 0.1% PBT overnight at 4°C. Samples were washed in 0.5% PBT, mounted in 90% glycerol in 1× PBS, and imaged within 1–4 days. Antibodies were obtained and used as follows: 1:300 Sox2 (R&D AF2018, RRID:AB_355110; neural stem cells), 1:100 12/101 (DSHB, RRID:AB_531892; skeletal muscle), and 1:400 phospho-Serine10-histone-H3 (P-H3; Millipore 06-570, RRID:AB_310177, mitotic marker). All donkey secondary antibodies were used at 1:1500–1:2000 from Thermo Fisher: anti-Goat-Alexa-647 (A21447, RRID:AB_141844), anti-Goat-Alexa-594 (A11058, RRID:AB_2534105), anti-Mouse-Alexa-488 (A21202, RRID:AB_141607), anti-Rabbit-Alexa-594 (A21207, RRID:AB_141637), and anti-Rabbit-Alexa-647 (A31573, RRID:AB_2536183). Phospho-CREB (P-CREB) immunostaining required conditions as follows: samples were fixed as above for 3 h at 4°C with gentle agitation, then washed in 0.1% PBT, dehydrated in methanol, and kept at −20°C overnight in methanol. Samples were then bleached at room temperature for 3 hr, rehydrated, permeabilized as above, blocked in 10% BSA + 1.5% normal donkey serum, and placed in primary antibody solution: 1:400 Sox2, 1:100 12/101, and 1:1500 P-CREB (Cell Signaling 9198) for 4–5 days at 4°C, then treated as above for secondary antibody incubation, washes, and mounting.

### Immunostaining in tissue sections

Larvae were fixed in 4% PFA for 1 h at room temperature. Samples were then processed for paraffin embedding and sectioned transversely in 10-µm-thick sections. Slides were processed for immunostaining by incubating overnight with the primary antibodies Sox2 (1:300) and 5E1 (1:50, recognizes *X. laevis* Shh and Ihh) at 4°C and for 2 h at room temperature with AlexaFluor conjugated secondary antibodies (1:300) in 1% BSA, 0.1% Tween in PBS. Immunostained samples were mounted and imaged in an epifluorescence microscope.

### Gli activity reporter

p8xGli-EGFP_CMV-iRFP670 was constructed as follows: the 8xGli-EGFP reporter plasmid was a gift from Prof. James Chen of Stanford University (*Sasaki et al., 1997*). The 8xGli-δcrystallin promoter was removed and spliced into the pXreg4-FireflyLuciferase_keratin-EGFP_CMV-RenillaLuciferase (courtesy of Dr. Yesser Hadj Balgacem, UC Davis, *Belgacem and Borodinsky, 2015*) in place of the

Xreg4-FL_keratin cassette to make p8xGli-EGFP_CMV-RenillaLuciferase. iRFP670 was then PCR-amplified from iRFP670-N1 (*Shcherbakova and Verkhusha, 2013*, Addgene 45457) and swapped with Renilla Luciferase to make p8xGli-EGFP_CMV-RFP670. 150–200 pg p8xGli-GFP_CMV-iRFP670 were injected per embryo at the 2–4 cell stage. Individual cells were selected in Imaris (Bitplane) from greater than twice background intensity iRFP670+ cells and analyzed for mean EGFP and iRFP670 intensity. The construct was validated in 18 h post-fertilization (hpf), neural plate stage embryos ± 10 nM SAG. Imaging took place at 18 or 60–88 hpf embryos or larvae, respectively. Neural plate stage validation is presented as scatter plots, by iRFP670 (X) and EGFP (Y) signal intensity, with all iRFP670+ cells from all samples pooled (*Figure 1—figure supplement 1*). Post-amputation Gli activity is presented as two-dimensional reconstructions of all iRFP670+ cells from all samples pooled, presented by each spot's relationship to the amputation plane (x axis) and the spinal cord (y axis), with each iRFP670+ cell assigned a heat map intensity by EGFP:iRFP670 ratio (*Figure 2*).

## Gli2 knockdown

Morpholino antisense oligonucleotides targeted to block Gli2 translation (*Belgacem and Borodinsky, 2015*, 5′-GCACAGAACGCAGGTAATGCTCCAT-3′, Gli2-MO) and matched Gli2-PhotoMO (5′-ATGGAGCATTACPTGCGTTCT-3′) were ordered from Gene Tools. During and after injection, all steps were performed in the dark, using only >580 nm light for illumination. Embryos were injected at 4-cell stage with 4 nl of solution containing a total of 2 pmol each Gli2-MO and PhotoMO blocker with Cascade Blue-Dextran tracer and grown at 20–22°C until 32 hpf. Larvae were then split into two separate Petri dishes in 10% MMR, and either kept in the dark (inactive control) or exposed to 30 min of 365 nm UV transillumination, low-output setting on a MaestroGen 240 V UV transilluminator (active MO). Larvae were then left at 20–22°C until 28 h post-UV, then amputated, and kept at 22–23°C until anesthesia, screening for tracer and fixation at 72 hpa.

## P-CREB+ cell quantification

### In whole mounts

Sox2 and P-CREB immunopositive cells within the spinal cord of immunostained whole mounts were separately analyzed on Imaris software (Bitplane). All cells were filtered for minimum 3× background Sox2 or P-CREB mean signal intensity, and background subtracted for mean Sox2 or P-CREB intensity. P-CREB+ cell counts are presented as the ratio of numbers of P-CREB+ cells per total number of Sox2+ nuclei for each 100 µm bin along the spinal cord posterior and anterior to the amputation plane for controls (data presented in *Figure 6B*), and as raw P-CREB count values for conditions treated with vehicle, SAG, or cyclopamine (data presented in *Figure 7D*).

### In sections

Larvae were amputated, incubated for 4 h with DMSO (control), 10 µM cyclopamine or 1 µM SAG, then fixed in 4% PFA for 1 h at room temperature. Samples were then processed for paraffin embedding and sectioned longitudinally in 10-µm-thick sections. Slides were processed for immunostaining by incubating overnight with the primary antibodies Sox2 (1:300), P-CREB (1:800) at 4°C and for 2 h at room temperature with AlexaFluor conjugated secondary antibodies (1:300) in 1% BSA, 0.1% Tween in PBS. Immunostained samples were mounted and imaged in an epifluorescence microscope. Number of P-CREB and Sox2 immunopositive cells was quantified by thresholding the signal to twice the background and normalized per 100 µm extension of the spinal cord, determined by the length of Sox2-labeled tissue. Data was collected considering as 0 µm the tip of the tail in 4-hpa larvae.

## Live and fixed sample imaging

Live larvae anesthetized with 0.02% TMS in 10% MMR were imaged under a Nikon swept-field confocal microscope using 488 nm (EGFP) and 647 nm (iRFP670) lasers. Fixed samples were imaged on Nikon-A1 or C2 point laser-scanning confocal microscopes using 488 nm (Alexa488), 561 nm (Alexa594), and 640 nm (Alexa647) lasers or using an Olympus epifluorescence microscope for immunostained sections.

## Western blot assay

Nuclear fraction was obtained from stage 39–40 Gli2MO + PhotoMO larvae (three larvae for each group) to assess endogenous expression of Gli2. Briefly, larvae injected with Gli2MO + PhotoMO +/− UV (as described above) were frozen in liquid nitrogen, stored at −80℃, then homogenized in 25 mM HEPES pH 7.4, 50 mM NaCl, 2 mM EGTA, 5 mM MgCl$_2$, protease inhibitors cocktail (784115, Thermo Fisher Scientific) on ice for 30 min and centrifuged for 10 min at 1000 g. Nuclear pellets were resuspended in 2× protein loading buffer (125 mM Tris-HCl, pH 6.8, 4% SDS, 20% (w/v) glycerol, 0.005% Bromophenol Blue, 5% β-mercaptoethanol) and boiled for 5 min. Samples were run in 10% SDS-PAGE and transferred to PVDF membrane. PVDF membrane was probed with anti-Gli2 goat polyclonal (AF3635; RRID:AB_211902), 1:800 in 5% BSA at 4℃, followed by incubation with horseradish peroxidase (HRP)-conjugated secondary antibody (711-035-152, Jackson ImmunoResearch; 1:10,000) and visualized by Western Lightning Plus-ECL, Enhanced Chemiluminescence Substrate (NEL103E001, Perkin Elmer). PVDF membranes were stripped in 0.2 M glycine HCl buffer, pH 2.5, 0.05% Tween for 20 min and re-probed with 1:500 anti-LaminII/III (Cell Signaling 9087; RRID:AB_10896336) for nucleus-specific loading control in 5% BSA.

For assessment of Hh ligand expression, whole-cell lysates were obtained from 500-μm-long stump/regenerate from 0 and 24 hpa larvae (35/group) by freezing samples in liquid nitrogen, then solubilizing in 1× Laemmli buffer and centrifuging for 5 min at 17,900 g. Supernatant containing 2.5% β-mercaptoethanol was boiled for 4 min. Samples were run in 4–20% gradient gel SDS-PAGE and transferred to PVDF membrane, which was probed with anti-Hh antibody 5E1 (DSBH), 1:50 in 5% BSA at 4℃ overnight, followed by incubation with HRP-conjugated goat anti-mouse secondary antibody (Millipore, 12-349) and visualized by ECL2. PVDF membranes were stripped in 0.2 M glycine HCl buffer, pH 2.5, 0.05% Tween for 20 min and re-probed with 1:1000 anti-GAPDH (SICGEN; RRID:AB_0049-200) for whole-cell loading control in 5% BSA, followed by incubation with HRP-conjugated rabbit anti-goat secondary antibody (R&D, HAF109).

All membranes were imaged with ChemiDoc-MP imaging instrument and optical density of bands of interest measured with associated software (Bio-Rad Laboratories).

## Experimental design and statistical analyses

All data were analyzed with Prism software (GraphPad). Data were first analyzed for normality, followed by parametric (normally distributed) or non-parametric tests (not normally distributed). When normally distributed and SDs were equal among groups, unpaired two-tail t-test or ordinary one-way ANOVA followed by Tukey's multiple comparisons test was used, when two or more groups were compared, respectively. When normally distributed and SDs were not equal among groups, Welch's t-test or Brown–Forsythe and Welch ANOVA followed by Dunnett's T3 multiple comparisons test was used, when two or more groups were compared, respectively. When data were not normally distributed, Mann–Whitney test or Kruskal–Wallis followed by Dunn's multiple comparisons test was used, when two or more groups were compared, respectively. Graphed values are presented as a normalized percent of the experiment-matched control treatment, pooled across experiments. Error bars for the pooled control groups coming from different experiments in *Figures 1*, *3,* and *4*, *Figure 4—figure supplement 1*, *Figures 5* and *7D* represent the largest error value for the control group in any of the experimental groups included in each graph. Significance was set to p<0.05. Number of samples and experiments is indicated in figure legends.

## Acknowledgements

We thank Dr. Yesser Hadj Belgacem for comments on the manuscript. This work was supported by NSF 1120796 and 1754340, NIH-NINDS R01NS073055, R01NS105886, R01NS113859, and Shriners Hospital for Children 86700-NCA grants to LNB and Shriners Hospital for Children Postdoctoral Fellowship to AMH. We thank Christopher Hom for assistance with data acquisition and analysis and Jacqueline Levin for technical advice and assistance.

## Additional information

### Funding

| Funder | Grant reference number | Author |
|---|---|---|
| National Science Foundation | 1120796 | Laura N Borodinsky |
| National Institute of Neurological Disorders and Stroke | R01NS073055 | Laura N Borodinsky |
| Shriners Hospitals for Children | 86700-NCA | Laura N Borodinsky |
| Shriners Hospitals for Children | 84303-NCA | Andrew M Hamilton |
| National Science Foundation | 1754340 | Laura N Borodinsky |
| National Institute of Neurological Disorders and Stroke | R01NS105886 | Laura N Borodinsky |
| National Institute of Neurological Disorders and Stroke | R01NS113859 | Laura N Borodinsky |

The funders had no role in study design, data collection and interpretation, or the decision to submit the work for publication.

### Author contributions
Andrew M Hamilton, Conceptualization, Data curation, Formal analysis, Funding acquisition, Investigation, Visualization, Methodology, Writing - original draft, Writing - review and editing; Olga A Balashova, Data curation, Formal analysis, Validation, Investigation, Methodology; Laura N Borodinsky, Conceptualization, Supervision, Funding acquisition, Writing - original draft, Project administration, Writing - review and editing

### Author ORCIDs
Laura N Borodinsky (iD) https://orcid.org/0000-0003-2937-7023

### Ethics
Animal experimentation: This study was performed in strict accordance with the recommendations in the Guide for the Care and Use of Laboratory Animals of the National Institutes of Health. All of the animals were handled according to approved institutional animal care and use committee (IACUC) protocols (#20537) of the University of California Davis. This institution is accredited by the Association for Assessment and Accreditation of Laboratory Animal Care, International (AAALAC). This institution has an Animal Welfare Assurance on file with the Office of Laboratory Animal Welfare (OLAW). The Assurance Number is A3433-01. All procedures were performed under tricaine anesthesia, and every effort was made to minimize suffering.

### Decision letter and Author response
Decision letter https://doi.org/10.7554/eLife.61804.sa1
Author response https://doi.org/10.7554/eLife.61804.sa2

## Additional files

### Supplementary files
• Transparent reporting form

### Data availability
All data generated or analysed during this study are included in the manuscript and supporting files.

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
