## [Decision Letter]

**Acceptance summary:**

This study shows that Hh signaling contributes to the regeneration of the *Xenopus* to spinal cord and muscle, after tail amputation at larva stages. The study confirms previous reports and further adds the notion that this function is mediated by a non-canonical branch of the pathway involving PKA and CREB underlies this process.

**Decision letter after peer review:**

Thank you for submitting your article "Non-canonical Hedgehog signaling regulates spinal cord and muscle regeneration" for consideration by *eLife*. Your article has been reviewed by 3 peer reviewers, one of whom is a member of our Board of Reviewing Editors, and the evaluation has been overseen by Marianne Bronner as the Senior Editor. The following individual involved in review of your submission has agreed to reveal their identity: Henry Roehl (Reviewer #2).

The reviewers have discussed the reviews with one another and the Reviewing Editor has drafted this decision to help you prepare a revised submission.

Summary:

Hh signalling contributes to the generation of a number of tissues. The present manuscript investigates how this signalling contributes to the regeneration of the *Xenopus* to spinal cord and muscle, after tail amputation at larva stages. The manuscript suggests that a non-canonical branch of the pathway involving PKA and CREB underlies this process.

The manuscript is well written and easy to follow. However the finding that distinguishes this manuscript from previous related reports -namely the involvement of non-canonical signalling in Hh-mediated regeneration- is poorly supported by the current data and needs to be strengthened by further experimentation.

Essential revisions:

1. The authors need to provide the rationale for using st 39-40 larvae instead older stages, commonly accepted as regenerative stages (i.e. st 49 to 54). This should be included in the method/results, further discussing the potential advantages/ disadvantages. More information on the timing of full regeneration at used stages should also be provided.

2. The authors should address the question of whether the activation of the non canonical pathway is ligand dependent or independent, among other by analysing whether HH ligands are in the initial 24 hr of regeneration.

3. The chemical inhibitors used in the study are not fully specific and therefore additional approaches should be used to validate the findings. One possibility is to use Photo-MO for Smoothened and CREB. Additional methods to support the involvement of PKA are also needed.

4. A direct link between pCREB and non-canonical HH signalling is needed. This could be provided by testing pCREB reduced after Smoothened inhibition during regeneration.

*Reviewer #1:*

This manuscript shows that spinal cord and muscle regeneration in the *Xenopus* larvae is mediated by non-canonical Hh signaling through the activity of PKA. This conclusion is based on a series of pharmacological treatments and a careful analysis of a number of quantified parameters.

The study adds to the idea that Hh signaling is important for tissue regeneration, with the additional interesting idea that the process is mediated, at least at early stages, by non-canonical signaling. The enthusiasm for these results is however undermined by a series of concerns:

According to the literature, st 49 to 54 are considered as the regenerative stages in *Xenopus laevis* (Edwards-Faret et al. 2017 Nat Protocols). Why have the authors used larvae at st 39-40? Can these earlier stages entail a contribution from developmental related aspects? How long does it take to reach full regeneration in these larvae? This information is missing and needs to be provided for non-specialists. This is important also because Figure 2 shows that Gli transcriptional activity begins to recover at 28 hr. When does it get back to normal levels? Is it before than 72 hrs (time used for most of the analysis)? Can Gli activity account for the differences observed between KT5720 and KT5720+ cyclop treatments?

On the same line: does treatment with GANT61 beginning 48hr (or so) after amputation has any effect on regeneration parameters?

The experiments reported in Figure 6 do not really prove that P-CREB mediates regeneration. Can the authors attempt to down-modulate its activity?

*Reviewer #2:*

In this manuscript the authors examine how Hh signalling is involved in spinal cord and muscle regeneration in the *Xenopus laevis*. The important finding is that a non-canonical branch of the Hedgehog pathway involving Protein Kinase A and CREB is acting early during *Xenopus* larval tail regeneration. The paper is well-written and the findings are very interesting, but further work is needed to warrant publication in *eLife*. We feel that these experiments fall within the authors expertise and could be accomplished within a reasonable timeframe.

1. Using the same regeneration model, Taniguchi et al. (2014) showed that all HH ligands become expressed in the notochord about 24 hours after injury. The analysis presented here is focused before this time. Is the hypothesis that non-canonical HH signalling is ligand independent? If so the Authors should confirm that HH ligands are not expressed during the period of their analysis (i.e. between 0 and 24 hours after injury).

2. The paper relies very heavily on chemical inhibitors which may not be entirely specific (see also point 3). The authors should consider using Photo-MO for Smothened to confirm some key results. This would be especially convincing when compared to the Gli MO (which did not have a strong effect).

3. KT5720 is used as an inhibitor of PKA. There are reports of potential off target effects including inhibiting MAPK (Pharmacological PKA Inhibition: All May Not Be What It Seems, Andrew J. Murray, 2008). MAPK can act upstream of CREB too and is known to be activated immediately after wounding (Christen and Slack, 1999). Could another method be used to strengthen the PKA hypothesis?

4. The Authors use pCREB as a read-out for non-canonical HH signalling in Figure 6. They need to show that pCREB is reduced by Smoothened inhibition during regeneration. What is the significance of its presence in skin after wounding, is that also dependent upon non-canonical HH signalling?

*Reviewer #3:*

This manuscript demonstrated the role of non-canonical Hh signaling in spinal cord and muscle regeneration in *Xenopus* tadpoles. The downstream effectors of this non-canonical Hh signaling involves PKA and CREB activation. The finding provides an explanation of the mechanisms of Hh signaling mediated tissue regeneration. Thus, it is of some interest for regenerative biology.

I have two concerns about this study.

First, for tadpole tail amputations, the authors used NF stages 39-40 embryos. While there is no consensus in the community on the *Xenopus* tadpole stages used for regeneration analysis, the tail buds of stage 39-40 embryos are not typically considered as differentiated tail, and the cells in the tail buds are precursor cells, rather than differentiated tissue cells. This is particularly true for the cells in the spinal cord. As has been well established, there is ontogenetic decline in the regeneration capacity of *Xenopus* tadpoles. Thus the stages of tadpoles used for tail amputation is an important concern when performing regeneration analysis.

Second, for Gli activity reporting, the authors injected EGFP:iRFP670 plasmid in the early embryos. This is not ideal, as transgenic tadpoles can be easily generated now.

[Editors' note: further revisions were suggested prior to acceptance, as described below.]

Thank you for submitting your article "Non-canonical Hedgehog signaling regulates spinal cord and muscle regeneration in *Xenopus laevis* larvae" for consideration by *eLife*. Your article has been reviewed by 3 peer reviewers, one of whom is a member of our Board of Reviewing Editors, and the evaluation has been overseen by Marianne Bronner as the Senior Editor. The following individual involved in review of your submission has agreed to reveal their identity: Henry Roehl (Reviewer #2).

The manuscript is much improved but there are a couple of key issues that still need to be carefully addressed and included in the manuscript to assure its publication in *eLife*.

Essential revisions:

1. The first relates to the question of whether Hh ligand expression is activated upon amputation at stage 40. Staining of Hh ligands needs to be analyzed and included in the manuscript. This is an important support for the model presented in Figure 9.

2. Graphs should be better presented. Error bars for controls are not treated consistently.

3. The method section related to how regeneration is measured in Figure 1—figure supplement should be expanded and include an explanation on what is the difference between the data presentation of Figure 1 and figure supplement 1.

*Reviewer #1:*

The manuscript has been revised addressing the comments raise during the first round of review. The manuscript will add new information of the role of Hh signalling in regeneration

*Reviewer #2:*

The responses to the reviewers' comments are on the whole satisfactory, except around the question of whether there is evidence that Hedgehog ligand expression is activated by excision at stage 40.

The authors state that "Taniguchi et al. reported an upregulation in shh expression upon amputation in the regenerating tail, but they do not argue that Hh is not expressed prior to amputation or during the first 24 hours, nor do their findings support this position." This is not accurate. Figure 1A of Taniguchi et al. analyses expression of shh and bhh over time. At 0hrs post amputation expression of shh is barely detectable, and bhh is not detected. At 24hrs post amputation there is no expression of either ligand. Starting at 48 hours post amputation, shh is induced strongly and bhh to a lesser degree. This expression is stated as being exclusively in the notochord. This has led to the model that Hedgehog signalling becomes slowly activated peaking four to five days after amputation. The authors should familiarise themselves with the literature on the role of Hedgehog signalling during tail regeneration. The comparison with other published analysis of tail regeneration is important even if it is at a different stage.

The authors show a staining of Hedgehog ligands in their rebuttal but do not present analysis of this data and have not added the data to their manuscript. It appears to show that Hedgehog ligands are strongly upregulated within 4 hours after amputation, and that this activation is primarily in the neural tube (not notochord?). This finding would be consistent with the author's model that Hedgehog signalling is activated by wounding and stimulates muscle and neural tube regeneration (Figure 9). Importantly this data would indicate that Hedgehog expression becomes active much earlier than has been previously described for stage 49. This data should be properly analysed and presented in the manuscript. It would provide important support for their model. This is not a labour intensive request.

*Reviewer #3:*

The revised manuscript added new data from additional experiment, explained the reasoning of using stage 39-40 tadpoles for tail amputation experiment, and replaced the western blot images showing the efficiency of *Gli2*-mo. The revision adequately addressed my concerns.

However, I find it confusing in understanding the graph presentations, especially with the new data. Error bars for controls are not treated consistently. For example, in Figure 1D-I, there are no error bars shown for the control groups, but in Figure 1—figure supplement 1 B-G, there are error bars shown for the control. Yet, the means for the controls in Figure 1—figure supplement 1 B-G are all 100. Could you please explain a bit more in the method section, how the regeneration is measured in Figure 1—figure supplement? What is the difference between the data presentation of Figure 1 and figure supplement 1?

---

## [Author Response]

Essential revisions:1. The authors need to provide the rationale for using st 39-40 larvae instead older stages, commonly accepted as regenerative stages (i.e. st 49 to 54). This should be included in the method/results, further discussing the potential advantages/ disadvantages. More information on the timing of full regeneration at used stages should also be provided.

A review of the literature on studies using *Xenopus* larvae as a model for tail regeneration results in a total of 63 original research articles, with 37 in which larvae of stages 39-42 are used and 26 in which stages older than 48 are used. Hence, the earlier stages that we used in this study are well accepted in the field as a good model for tail regeneration in this system. The reasoning behind this is that unlike the tip of the tail that is still developing and growing, the tissues of the stump that become exposed after amputation correspond to mature larval tissues, thus the replenishing of these tissues represents a regenerative process and not the mere developmental progression of tissue growth.

Nevertheless, this is an important point that we are thankful the reviewers brought to our attention so that we can better introduce the readers to this model system. We revised the Results section accordingly (Page 5, new paragraph 1).

2. The authors should address the question of whether the activation of the non canonical pathway is ligand dependent or independent, among other by analysing whether HH ligands are in the initial 24 hr of regeneration.

Expression of Shh has been documented during these larval stages mostly through *in situ* hybridization assays (Howell et al., 2002; Koide et al., 2006; Yin et al., 2010; Ekker et al., 1995, Kazanskaya et al. 2000). From the 4 hedgehog genes identified in *Xenopus laevis*, Shh, expressed in notochord and/or floor plate, and banded hedgehog (bhh), expressed in somites (Ekker et al., 1995), are likely the potential ligands of Patched that activate Smo signaling during regeneration at these larval stages. Moreover, immunostaining of transverse sections of non-amputated and 4 hpa-amputated larvae shows immunolabeling for Hh (5E1 antibody that detects both *Xenopus* shh and bhh, Developmental Studies Hybridoma Bank) in somites of larvae at these stages and in spinal cord of regions proximal to the amputation plane (see Author response image 1).

We revised the manuscript to include references that report expression of Hh ligands at these stages of development (page 5, last paragraph).

3. The chemical inhibitors used in the study are not fully specific and therefore additional approaches should be used to validate the findings. One possibility is to use Photo-MO for Smoothened and CREB. Additional methods to support the involvement of PKA are also needed.

Cyclopamine and SAG have been extensively validated and utilized as both highly specific and efficient antagonist and agonist respectively of Smoothened, Hh signaling major effector. The main focus of this study is to demonstrate the necessity for non-canonical Hh, Smoothened-dependent, Gli-independent signaling for muscle and spinal cord regeneration. Hence, we complemented the pharmacological loss and gain of Smoothened function approaches with both pharmacological and genetic loss of Gli function by treating amputated larvae with GANT61, Gli transcriptional activity inhibitor, and by microinjecting *Gli2*-photo-morpholino. In addition, we also report the decrease in Gli activity upon amputation. We employed multiple approaches to demonstrate the main scope of this study.

Nevertheless, we performed additional experiments that further support the conclusion of Smoothened-dependent regeneration of muscle and spinal cord by incubating amputated larvae with another highly specific Smoothened inhibitor, vismodegib, that binds to a different domain than cyclopamine, and obtained similar effects on regeneration of muscle and spinal cord as with cyclopamine. These new data are included as a new supplement to Figure 1 in the revised manuscript (Figure 1 – supplement 1) and described in the Results section (page 6, second paragraph, lines 101-104).

We chose pharmacological means to manipulate signaling because this study requires temporal control so that the inhibition of the signaling is restricted to the regeneration period. In addition, Gene Tools, the company that develops morpholinos, has discontinued the production of photo-morpholinos. We have attempted to utilize Smo *vivo*-morpholino, that, unlike the regular morpholinos, permeate into cells, thus, do not need to be injected in the fertilized embryo. However, high variability in regeneration metrics observed in pilot experiments and difficulty in successfully staining for Smo in whole mounts led to inconsistent results and uncertainty of the efficiency of the intended manipulation.

We agree with the reviewers that further investigation is needed to identify the non-canonical Smo-dependent signaling involved in the regeneration of muscle and spinal cord. However, we consider this belongs to a follow up study and believe the findings presented in this manuscript contribute with significant and relevant understanding on Hh-dependent tissue regeneration.

4. A direct link between pCREB and non-canonical HH signalling is needed. This could be provided by testing pCREB reduced after Smoothened inhibition during regeneration.

We have performed additional experiments to directly address the link between the non-canonical HH signaling and pCREB during regeneration by incubating amputated larvae in the presence or absence of SAG or cyclopamine followed by immunostaining of samples for pCREB. Results show that cyclopamine or SAG incubation for the first 4 hpa respectively decreases or increases the number of pCREB+ nuclei in the spinal cord proximal to the tip of the amputated tail. These results suggest that HH signaling modulates CREB activity in the regenerating spinal cord. Moreover, we find that 24-h incubation with cyclopamine or SAG decreases or increases respectively the number of pCREB immunopositive nuclei in the regenerating spinal cord. We revised the manuscript to include these new sets of data (Results, page 12, last paragraph; Figure 7, in current revised manuscript).

Reviewer #1:This manuscript shows that spinal cord and muscle regeneration in the *Xenopus larvae* is mediated by non-canonical Hh signaling through the activity of PKA. This conclusion is based on a series of pharmacological treatments and a careful analysis of a number of quantified parameters.The study adds to the idea that Hh signaling is important for tissue regeneration, with the additional interesting idea that the process is mediated, at least at early stages, by non-canonical signaling. The enthusiasm for these results is however undermined by a series of concerns:According to the literature, st 49 to 54 are considered as the regenerative stages in *Xenopus laevis* (Edwards-Faret et al. 2017 Nat Protocols). Why have the authors used larvae at st 39-40? Can these earlier stages entail a contribution from developmental related aspects? How long does it take to reach full regeneration in these larvae? This information is missing and needs to be provided for non-specialists.

A review of the literature on studies using *Xenopus* larvae as a model for tail regeneration results in a total of 63 original research articles, with 37 in which larvae of stages 39-42 are used and 26 in which stages older than 48 are used. Hence, the earlier stages that we used in this study are well accepted in the field as a good model for tail regeneration in this system. The reasoning behind this is that unlike the tip of the tail that is still developing and growing, the tissues of the stump that become exposed after amputation correspond to mature larval tissues, thus the replenishing of these tissues represents a regenerative process and not the mere developmental progression of tissue growth.

Nevertheless, this is an important point that we are thankful the reviewers brought to our attention so that we can better introduce the readers to this model system. We revised the Results section accordingly (Page 5, new paragraph 1).

Studies have shown that full regeneration is achieved after 7-10 days post tail amputation in these larval stages. Nonetheless, the current study is focused on the first stages of regeneration and the signaling mechanisms responding to the acute injury. Hence, the latest end point for the assays presented is 3 days post amputation, when regeneration of muscle and spinal cord are appreciable and measurements of extent of regeneration are reproducible, to allow comparison across controls and experimental groups.

This is important also because Figure 2 shows that Gli transcriptional activity begins to recover at 28 hr. When does it get back to normal levels? Is it before than 72 hrs (time used for most of the analysis)? Can Gli activity account for the differences observed between KT5720 and KT5720+ cyclop treatments?

Because cyclopamine effect in 24-h only group is already significant, this suggests that Smo activity during these first 24 h is important. In contrast, GANT61 and *Gli2*-MO incubated for the full 72 h post amputation are almost inactive upon the process of spinal cord and muscle regeneration. Hence, we conclude that the canonical, Gli-dependent Hh signaling does not contribute significantly to this process during the first 3 days of regeneration.

We do not think that Gli activity explains the differences in regeneration between inhibiting PKA alone or along with Smo inhibition because inhibiting Gli directly either pharmacologically with GANT61 or genetically by activating *Gli2*-MO does not result in any appreciable effect on regeneration of muscle or spinal cord. Instead, we think that other non-canonical Smo-dependent signaling, independent of PKA is triggered; and conversely some PKA-dependent regeneration may operate independently from Smo pathway.

On the same line: does treatment with GANT61 beginning 48hr (or so) after amputation has any effect on regeneration parameters?

Treatments for 72 h were done so that the drugs were replenished in the saline that larvae were incubated in every day. Hence, the 72-h treatment implemented for GANT61 consisted on replenishing the solution with GANT61 each day of the 3-day period. Any effect of inhibiting Gli1/2 transcriptional activity during the last day of regeneration (48-72 h) should have been observed. We interpret the lack of significant effect on almost every parameter of regeneration as a lack of participation/necessity of Hh Gli-dependent canonical pathway in the early stages (first 3 days) of the regenerative process.

We do not expect a different result from starting incubation with GANT61 at 48 hpa, compared to 0 hpa.

The experiments reported in Figure 6 do not really prove that P-CREB mediates regeneration. Can the authors attempt to down-modulate its activity?

We believe that inhibiting PKA inhibits CREB activation, as we and many others have demonstrated in previous studies (Belgacem and Borodinsky 2015). Thus, the effect of PKA inhibitor on spinal cord and muscle regeneration may be revealing a role for P-CREB on the regeneration of these tissues.

We agree with the reviewers that assessing more specifically the role of P-CREB will be an interesting future direction for this investigation. However, we believe it belongs to a follow-up study beyond the scope of the present manuscript.

Reviewer #2:In this manuscript the authors examine how Hh signalling is involved in spinal cord and muscle regeneration in the *Xenopus laevis*. The important finding is that a non-canonical branch of the Hedgehog pathway involving Protein Kinase A and CREB is acting early during *Xenopus larval* tail regeneration. The paper is well-written and the findings are very interesting, but further work is needed to warrant publication in eLife. We feel that these experiments fall within the authors expertise and could be accomplished within a reasonable timeframe.1. Using the same regeneration model, Taniguchi et al. (2014) showed that all HH ligands become expressed in the notochord about 24 hours after injury. The analysis presented here is focused before this time. Is the hypothesis that non-canonical HH signalling is ligand independent? If so the Authors should confirm that HH ligands are not expressed during the period of their analysis (i.e. between 0 and 24 hours after injury).

Taniguchi et al. (2014) used stage 49 tadpoles (12 day-old) instead of stage 40 larvae (2.75 day-old) as in this study. Hence it is difficult to compare. Nevertheless, Taniguchi et al. reported an upregulation in shh expression upon amputation in the regenerating tail, but they do not argue that Hh is not expressed prior to amputation or during the first 24 hours, nor do their findings support this position.

In addition, expression of Shh has been documented during these larval stages mostly through *in situ* hybridization assays (Howell et al., 2002; Koide et al., 2006; Yin et al., 2010; Ekker et al., 1995, Kazanskaya et al. 2000). From the 4 hedgehog genes identified in *Xenopus laevis*, Shh, expressed in notochord and/or floor plate, and banded hedgehog (bhh), expressed in somites (Ekker et al., 1995), are likely the potential ligands of Patched that activate Smo signaling during regeneration at these larval stages. Moreover, immunostaining of transverse sections of non-amputated and 4 hpa-amputated larvae shows immunolabeling for Hh (5E1 antibody that detects both *Xenopus* shh and bhh, Developmental Studies Hybridoma Bank) in somites of larvae at these stages and in spinal cord of regions proximal to the amputation plane (see Author response image 1).

We revised the manuscript to include references that report expression of Hh ligands at these stages of development (page 5, last paragraph).

2. The paper relies very heavily on chemical inhibitors which may not be entirely specific (see also point 3). The authors should consider using Photo-MO for Smothened to confirm some key results. This would be especially convincing when compared to the Gli MO (which did not have a strong effect).

Every approach has its pros and cons. The pharmacological approach offers a higher temporal resolution of the down or upregulation of a signaling pathway, which is particularly important in this case when we want to impose the loss or gain of function of a certain pathway strictly during injury and regeneration.

We agree with the reviewer that showing the same outcome with multiple approaches strengthens the relevance and significance of the finding. Nevertheless, Cyclopamine and SAG have been extensively validated and utilized as both highly specific and efficient antagonist and agonist respectively of Smoothened, Hh signaling major effector. The main focus of this study is to demonstrate the necessity for non-canonical Hh, Smoothened-dependent, Gli-independent signaling for muscle and spinal cord regeneration. Hence, we complemented the pharmacological loss and gain of Smoothened function approaches with both pharmacological and genetic loss of Gli function by treating amputated larvae with GANT61, Gli transcriptional activity inhibitor, and by microinjecting *Gli2*-photo-morpholino. In addition, we also report the decrease in Gli activity upon amputation. We employed multiple approaches to demonstrate the main scope of this study.

We also performed additional experiments that further support the conclusion of Smoothened-dependent regeneration of muscle and spinal cord by incubating amputated larvae with another highly specific Smoothened inhibitor, vismodegib, that binds to a different domain than cyclopamine, and obtained similar effects on regeneration of muscle and spinal cord as with cyclopamine. These new data are included as a new supplement to Figure 1 in the revised manuscript (Figure 1 – supplement 1) and described in the Results section (page 6, second paragraph, lines 101-104).

In addition, Gene Tools, the company that develops morpholinos, has discontinued the production of photo-morpholinos. We have attempted to utilize Smo *vivo*-morpholino, that, unlike the regular morpholinos, permeate into cells, and thus do not need to be injected. However, high variability in regeneration metrics observed in pilot experiments and difficulty in successfully staining for Smo in whole mounts led to inconsistent results and uncertainty of the efficiency of the intended manipulation.

3. KT5720 is used as an inhibitor of PKA. There are reports of potential off target effects including inhibiting MAPK (Pharmacological PKA Inhibition: All May Not Be What It Seems, Andrew J. Murray, 2008). MAPK can act upstream of CREB too and is known to be activated immediately after wounding (Christen and Slack, 1999). Could another method be used to strengthen the PKA hypothesis?

We agree with the reviewer and appreciate the reviewer’s suggestion of strengthening the results from pharmacologically inhibiting PKA. However, the main focus of this manuscript is to demonstrate that a non-canonical, Hh signaling is necessary for spinal and muscle regeneration, while the canonical Gli-dependent signaling is dispensable and suppressed during the early stages of tissue regeneration.

We believe we convincingly demonstrate that and provide further evidence of potential partners in the Hh non-canonical signaling without being exhaustive or definitive about the identity of these signaling mechanisms.

Although we used in previous studies overexpression of dominant negative and constitutively active forms of PKA, by microinjecting these constructs in 2-cell stage embryos (Belgacem and Borodinsky, 2011), for the current study focused on later larval stages and restricted to the regenerative process after tail amputation, this approach is not adequate because it interferes with embryonic development.

We present the inhibition of PKA as a potential candidate pathway, but we agree with the reviewer that other pathways may be affected by the PKA inhibitor and participating in the regeneration of muscle and spinal cord. Further investigation is needed and will be pursued in a follow up study. We have revised the manuscript to indicate that the drug may affect other kinases that may participate in this process, that are also known to activate CREB (page 11, lines 217-220).

Nonetheless, we now provide further evidence in the revised manuscript of Smo-dependent activation of P-CREB in the spinal cord proximal to the site of injury (Figure 7), strengthening the link between Smo and the non-canonical signaling, similar to what we have identified in the embryonic spinal cord (Belgacem and Borodinsky, 2015).

4. The Authors use pCREB as a read-out for non-canonical HH signalling in Figure 6. They need to show that pCREB is reduced by Smoothened inhibition during regeneration. What is the significance of its presence in skin after wounding, is that also dependent upon non-canonical HH signalling?

We have performed additional experiments to directly address the link between the non-canonical HH signaling and pCREB during regeneration by incubating amputated larvae in the presence or absence of SAG or cyclopamine followed by immunostaining of samples for pCREB. Results show that cyclopamine or SAG incubation for the first 4 hpa respectively decreases or increases the number of pCREB+ nuclei in the spinal cord proximal to the tip of the amputated tail. These results suggest that HH signaling modulates CREB activity in the regenerating spinal cord. Moreover, we find that 24-h incubation with cyclopamine or SAG decreases or increases respectively the number of pCREB immunopositive nuclei in the regenerating spinal cord. We revised the manuscript to include these new sets of data (Results, page 12, last paragraph; Figure 7, in current revised manuscript).

Reviewer #3:This manuscript demonstrated the role of non-canonical Hh signaling in spinal cord and muscle regeneration in *Xenopus tadpoles*. The downstream effectors of this non-canonical Hh signaling involves PKA and CREB activation. The finding provides an explanation of the mechanisms of Hh signaling mediated tissue regeneration. Thus, it is of some interest for regenerative biology.I have two concerns about this study.First, for tadpole tail amputations, the authors used NF stages 39-40 embryos. While there is no consensus in the community on the *Xenopus tadpole* stages used for regeneration analysis, the tail buds of stage 39-40 embryos are not typically considered as differentiated tail, and the cells in the tail buds are precursor cells, rather than differentiated tissue cells. This is particularly true for the cells in the spinal cord. As has been well established, there is ontogenetic decline in the regeneration capacity of *Xenopus tadpoles*. Thus the stages of tadpoles used for tail amputation is an important concern when performing regeneration analysis.

A review of the literature on studies using *Xenopus* larvae as a model for tail regeneration results in a total of 63 original research articles, with 37 in which larvae of stages 39-42 are used and 26 in which stages older than 48 are used. Hence, the earlier stages that we used in this study are well accepted in the field as a good model for tail regeneration in this system. The reasoning behind this is that unlike the tip of the tail that is still developing and growing, the tissues of the stump that become exposed after amputation correspond to mature larval tissues, thus the replenishing of these tissues represents a regenerative process and not the mere developmental progression of tissue growth.

Nevertheless, this is an important point that we are thankful the reviewers brought to our attention so that we can better introduce the readers to this model system. We revised the Results section accordingly (Page 5, new paragraph 1).

Second, for Gli activity reporting, the authors injected EGFP:iRFP670 plasmid in the early embryos. This is not ideal, as transgenic tadpoles can be easily generated now.

We appreciate the reviewer’s suggestion. However, we consider that the mosaic expression of the reporter actually plays to our advantage in allowing us to better assess in sparsely labeled cells Gli activity through live imaging. If reporter expression is extended to all cells as in transgenic larvae, the ability to distinguish between individual cells would have been difficult, if not impossible. Additionally, transgenesis is not an established methodology in the lab and currently we do not have the capability of breeding transgenic animals. Moreover, although transient F0 transgenic larvae would be doable in the lab, the efficiency of transgenesis for big constructs is low and the reporter we developed for this study is 6,378 bp, on the large side of constructs.

[Editors' note: further revisions were suggested prior to acceptance, as described below.]

The manuscript is much improved but there are a couple of key issues that still need to be carefully addressed and included in the manuscript to assure its publication in eLife.Essential revisions:1. The first relates to the question of whether Hh ligand expression is activated upon amputation at stage 40. Staining of Hh ligands needs to be analyzed and included in the manuscript. This is an important support for the model presented in Figure 9.

To assess Hh ligand protein expression upon amputation, we performed Western blot assays from 500 mm-long regions of the tail stump at 0 and 24 h after amputation. We probed membranes with whole-cell lysates from these samples with the Hh 5E1 antibody (DSHB). Results show that Hh ligands are detected before amputation and during early regeneration. Moreover, we find an increase in Hh protein levels in regenerating tail after 24 h post amputation compared to 0 h. We included these additional results in a new Figure 1- supplement 1, along with an image of 4 hours post amputation larvae stained with the 5E1 antibody to illustrate the spatial distribution of Hh ligands during the first hours post amputation.

We attempted to do a quantitative analysis of 5E1 immunostaining in tissue sections of non-amputated compared to amputated larvae, but we were not satisfied with the reliability of this approach. Due to the pattern of the staining, which mostly localizes to cell membrane, we were not able to count cells or threshold the signal in a consistent manner across samples. Nevertheless, data on the expression of Hh ligand transcripts in the same model system at the same developmental and post amputation times that we used for our study have been recently published using single-cell RNA seq (Aztekin et al., Science 2019). We browsed the data shared (https://marionilab.cruk.cam.ac.uk/XenopusRegeneration/) by the authors and found that Hh ligands are indeed expressed in stage 40 larvae. Transcripts for shh are detected in posterior notochord and floor plate in pre-amputated and in 1 through 3 days post amputation of the regenerating tail, with an increase in levels of shh transcript as early as 1 day post amputation, which is the earliest time point presented by the authors. Transcripts for shh are also expressed in neural tissue, including spinal cord progenitors and subtypes of spinal cord neurons, while dhh is detected in neural tissue and floor plate.

It should be noted that the pattern of immunohistochemistry that we included in our prior responses to reviewers and in the new Figure 1- supplement 1 may detect not only cells/tissues that express the Hh ligands but also target cells that receive it. Indeed, Aztekin and colleagues reported expression of ptch1 and ptch2 genes in the myotome, neural tissue and floor plate.

2. Graphs should be better presented. Error bars for controls are not treated consistently.

We have revised graphs in Figures 1, 3, 4, 5 and 7D to keep a consistent presentation of error bars for control groups.

3. The method section related to how regeneration is measured in Figure 1—figure supplement should be expanded and include an explanation on what is the difference between the data presentation of Figure 1 and figure supplement 1.

We have revised the presentation of the data so that is consistent across figures and figure supplements and revised the text in the Methods section (page 27, first paragraph) to explain how control groups and error bars were treated for each set of data.

Reviewer #2:The responses to the reviewers' comments are on the whole satisfactory, except around the question of whether there is evidence that Hedgehog ligand expression is activated by excision at stage 40.The authors state that "Taniguchi et al. reported an upregulation in shh expression upon amputation in the regenerating tail, but they do not argue that Hh is not expressed prior to amputation or during the first 24 hours, nor do their findings support this position." This is not accurate. Figure 1A of Taniguchi et al. analyses expression of shh and bhh over time. At 0hrs post amputation expression of shh is barely detectable, and bhh is not detected. At 24hrs post amputation there is no expression of either ligand. Starting at 48 hours post amputation, shh is induced strongly and bhh to a lesser degree. This expression is stated as being exclusively in the notochord. This has led to the model that Hedgehog signalling becomes slowly activated peaking four to five days after amputation. The authors should familiarise themselves with the literature on the role of Hedgehog signalling during tail regeneration. The comparison with other published analysis of tail regeneration is important even if it is at a different stage.

We have revised the text to include more recent studies that have addressed the expression of Hh ligands during regeneration of the tail at the developmental stages used in this study (Results section, page 5 last sentence and page 6 first paragraph). We browsed the single-cell RNA seq data shared (https://marionilab.cruk.cam.ac.uk/XenopusRegeneration/) by Aztekin et al. (Science 2019), given that they used *X. laevis* larvae at the same developmental and post amputation times that we used for our study, and found that Hh ligands are indeed expressed in stage 40 larvae. Transcripts for shh are detected in posterior notochord and floor plate in pre-amputated and in 1 through 3 days post amputation of the regenerating tail, with an increase in levels of shh transcript as early as 1 day post amputation, which is the earliest time point presented by the authors. Transcripts for shh are also expressed in the neural tissue, including spinal cord progenitors and subtypes of spinal cord neurons, while dhh is detected in neural tissue and floor plate.

Comparing patterns of expression in the regenerating tail between stage 40 and stage 49 larvae is problematic, and it is not that surprising that there are differences between the findings of Taniguchi et al. in stage 49 larvae and Aztekin et al. in stage 40 animals. It is also important to consider the methodological approaches used to assess expression, which exhibit different sensitivities and spatiotemporal resolution of expression levels. In addition, these studies report on Hh ligand transcript expression; whether transcripts correlate directly with protein levels is doubtful. For instance, Author response image 2 shows a representation of shh mRNA and protein from Xenbase, where the developmental changes in transcript and protein levels strikingly diverge. This suggests that reporting on transcript levels of a signaling molecule may not predict the temporal course of activation of a certain signaling pathway.

**Author response image 2. respfig2:** *X. laevis* embryonic protein expression. This graph shows protein expression in *Xenopus laevis*. Optionally the mRNA expression is also charted for comparison. The number in brackets after the gene symbol indicates the number of peptides. (Peshkin et al., 2019)

The authors show a staining of Hedgehog ligands in their rebuttal but do not present analysis of this data and have not added the data to their manuscript. It appears to show that Hedgehog ligands are strongly upregulated within 4 hours after amputation, and that this activation is primarily in the neural tube (not notochord?). This finding would be consistent with the author's model that Hedgehog signalling is activated by wounding and stimulates muscle and neural tube regeneration (Figure 9). Importantly this data would indicate that Hedgehog expression becomes active much earlier than has been previously described for stage 49. This data should be properly analysed and presented in the manuscript. It would provide important support for their model. This is not a labour intensive request.

To assess Hh ligand protein expression upon amputation, we performed Western blot assays from 500 mm-long regions of the tail stump at 0 and 24 h after amputation. We probed membranes with whole-cell lysates from these samples with the Hh 5E1 antibody (DSHB). Results show that Hh ligands are detected before amputation and during early regeneration. Moreover, we find an increase in Hh protein levels in regenerating tail after 24 h post amputation compared to 0 h.

We included these additional results in new Figure 1- supplement 1, along with an image of 4 hours post amputation larvae stained with the 5E1 antibody to illustrate the spatial distribution of Hh ligands during the first hours post amputation.

We attempted to do a quantitative analysis of 5E1 immunostaining in tissue sections of non-amputated compared to amputated larvae, but we were not satisfied with the reliability of this approach. Due to the pattern of the staining, which mostly localizes to cell membrane, we were not able to count cells or threshold the signal in a consistent manner across samples. Nevertheless, data on the expression of Hh ligand transcripts in the same model system at the same developmental and post amputation times that we used for our study have been recently published using single-cell RNA seq (Aztekin et al., Science 2019). We browsed the data shared (https://marionilab.cruk.cam.ac.uk/XenopusRegeneration/) by the authors and found that Hh ligands are indeed expressed in stage 40 larvae. Transcripts for shh are detected in posterior notochord and floor plate in pre-amputated and in 1 through 3 days post amputation of the regenerating tail, with an increase in level of shh transcript as early as 1 day post amputation, which is the earliest time point presented by the authors. Transcripts for shh are also expressed in neural tissue, including spinal cord progenitors and subtypes of spinal cord neurons, while dhh is detected in neural tissue and floor plate.

It should be noted that the pattern of immunohistochemistry that we included in our prior responses to reviewers and in the new Figure 1-supplement 1 may detect not only cells/tissues that express the Hh ligands but also target cells that receive it. Indeed, Aztekin and colleagues reported expression of ptch1 and ptch2 genes in the myotome, neural tissue and floor plate.

Reviewer #3:The revised manuscript added new data from additional experiment, explained the reasoning of using stage 39-40 tadpoles for tail amputation experiment, and replaced the western blot images showing the efficiency of Gli2-mo. The revision adequately addressed my concerns.However, I find it confusing in understanding the graph presentations, especially with the new data. Error bars for controls are not treated consistently. For example, in Figure 1D-I, there are no error bars shown for the control groups, but in Figure 1—figure supplement 1 B-G, there are error bars shown for the control. Yet, the means for the controls in Figure 1—figure supplement 1 B-G are all 100. Could you please explain a bit more in the method section, how the regeneration is measured in Figure 1—figure supplement? What is the difference between the data presentation of Figure 1 and figure supplement 1?

Error bars in control groups were used in Figure 1 supplement 1 (current Figure 1-supplement 2 and several others) because the control conditions were perfectly cohort-matched with treatment conditions, such that the error bars for controls would apply exactly to the treatments depicted; i.e. all of the data that went into said bar graphs came from the same experimental cohorts. In contrast, the control bars in Figure 1 and others did not have error bars because some independent experimental sets were combined for presentation: i.e. the cohort-matched controls for each treatment bar were not the same samples. For example, some cyclopamine treatments did not have matched SAG treatments in their cohort, and hence the controls for the SAG and cyclopamine bars were not the same samples. All metrics presented were pooled for analysis as a percent of the average, cohort-paired control values for each parameter in question.

In the interest of consistency in the presentation of the data and to report on the variability inherent to control groups in all figures, we have added error bars to the controls corresponding to the largest standard error values in the control groups of the different independent experiments that were included in the given bar graph.

Hence, we revised graphs in Figures 1, 3, 4—figure supplement 1, 5 and 7 to keep a consistent presentation of error bars for control groups.

We revised the text in the Methods section (page 27, first paragraph) to explain how control groups and error bars were treated for each set of data presented.